# Scaling Laws For Deep Learning Based Image Reconstruction

**Tobit Klug & Reinhard Heckel**
Dept. of Computer Engineering
Technical University of Munich
Munich, Bavaria, Germany
`{tobit.klug,reinhard.heckel}@tum.de`

## Abstract

Deep neural networks trained end-to-end to map a measurement of a (noisy) image to a clean image perform excellent for a variety of linear inverse problems. Current methods are only trained on a few hundreds or thousands of images as opposed to the millions of examples deep networks are trained on in other domains. In this work, we study whether major performance gains are expected from scaling up the training set size. We consider image denoising, accelerated magnetic resonance imaging, and super-resolution and empirically determine the reconstruction quality as a function of training set size, while simultaneously scaling the network size. For all three tasks we find that an initially steep power-law scaling slows significantly already at moderate training set sizes. Interpolating those scaling laws suggests that even training on millions of images would not significantly improve performance. To understand the expected behavior, we analytically characterize the performance of a linear estimator learned with early stopped gradient descent. The result formalizes the intuition that once the error induced by learning the signal model is small relative to the error floor, more training examples do not improve performance.

## 1 Introduction

Deep neural networks trained to map a noisy measurement of an image or a noisy image to a clean image give state-of-the-art (SOTA) performance for image reconstruction problems. Examples are image denoising (Burger et al., 2012; Zhang et al., 2017; Brooks et al., 2019; Liang et al., 2021), super-resolution (Dong et al., 2016; Ledig et al., 2017; Liang et al., 2021), and compressive sensing for computed tomography (CT) (Jin et al., 2017), and accelerated magnetic resonance imaging (MRI) (Zbontar et al., 2018; Sriram et al., 2020; Muckley et al., 2021; Fabian & Soltanolkotabi, 2022).

The performance of a neural network for imaging is determined by the network architecture and optimization, the size of the network, and the size and quality of the training set.

Significant work has been invested in architecture development. For example, in the field of accelerated MRI, networks started out as convolutional neural networks (CNN) (Wang et al., 2016), which are now often used as building blocks in un-rolled variational networks (Hammernik et al., 2018; Sriram et al., 2020). Most recently, transformers have been adapted to image reconstruction (Lin & Heckel, 2022; Huang et al., 2022; Fabian & Soltanolkotabi, 2022).

However, it is not clear how substantial the latest improvements through architecture design are compared to potential improvements expected by scaling the training set and network size. Contrary to natural language processing (NLP) models and modern image classifiers that are trained on billions of examples, networks for image reconstruction are only trained on hundreds to thousands of example images. For example, the training set of SwinIR (Liang et al., 2021), the current SOTA for image denoising, contains only 10k images, and the popular benchmark dataset for accelerated MRI consists only of 35k images (Zbontar et al., 2018).

In this work, we study whether neural networks for image reconstruction only require moderate amounts of data to reach their peak performance, or whether major boosts are expected from increasing

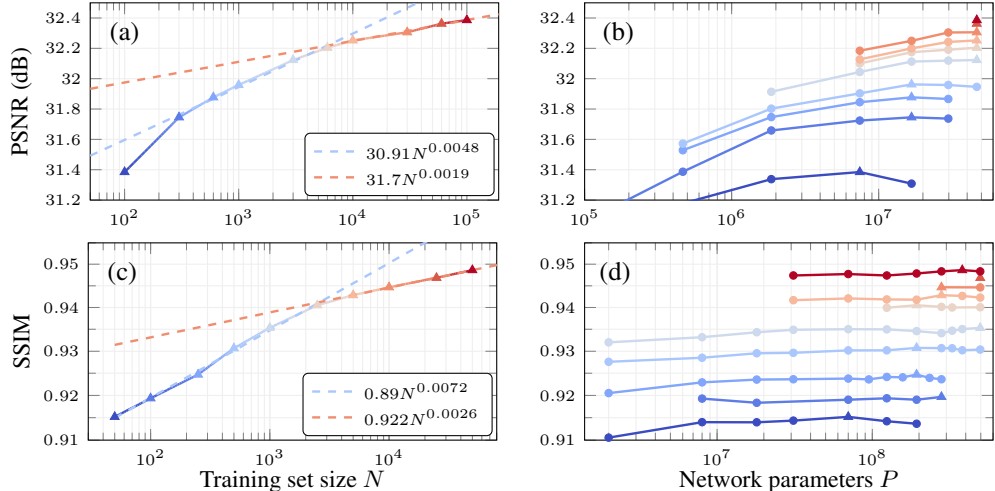

Figure 1: **Empirical scaling laws for CNN-based image reconstruction.** Reconstruction performance of a U-Net for denoising (a) and accelerated MRI (c) as a function of the training set size $N$. In both experiments an initial steep power law $R = \beta N^\alpha$ transitions to a relatively flat one already at moderate $N$. Thus we expect that training on millions of images does not significantly improve performance. To obtain the scaling curves in (a),(c) we optimize over the number of network parameters as shown in (b),(d). Colors in the plots on the left and right correspond to the same training set size. Since for large training set sizes corresponding to the flattened scaling law, increasing the parameters does not boost performance further, the decay in scaling coefficients is a robust finding.

the training set size. To partially address this question, we focus on three problems: image denoising, reconstruction from few and noisy measurements (compressive sensing) in the context of accelerated MRI, and super-resolution. We pick Gaussian denoising for its practical importance and since it can serve as a building block to solve more general image reconstruction problems well (Venkatakrishnan et al., 2013). We pick MR reconstruction because it is an important instance of a compressive sensing problem, and many problems can be formulated as compressive sensing problems, for example super-resolution and in-painting. In addition, for MR reconstruction the question on how much data is needed is particularly important, since it is expensive to collect medical data.

For the three problems we identify scaling laws that describe the reconstruction quality as a function of the training set size, while simultaneously scaling network sizes. Such scaling laws have been established for NLP and classification tasks, as discussed below, but not for image reconstruction.

The experiments are conducted with a U-Net (Ronneberger et al., 2015) and the SOTA SwinIR (Liang et al., 2021), a transformer architecture. We primarily consider the U-Net since it is widely used for image reconstruction and acts as a building block in SOTA models for image denoising (Brooks et al., 2019; Gurrola-Ramos et al., 2021; Zhang et al., 2021; 2022) and accelerated MRI (Zbontar et al., 2018; Sriram et al., 2020). We also present results for denoising with the SwinIR. The SwinIR outperforms the U-Net, but we find that its scaling with the number of training examples does not differ notably. Our contributions are as follows:

- **Empirical scaling laws for denoising.** We train U-Nets of sizes 0.1M to 46.5M parameters with training set sizes from 100 to 100k images from the ImageNet dataset (Russakovsky et al., 2015) for Gaussian denoising with 20.17dB Peak-Signal-to-Noise ratio (PSNR). While for the largest training set sizes and network sizes we consider, performance continues to increase, the rate of performance increase for training set sizes beyond a few thousand images slows to a level indicating that even training on millions of images only yields a marginal benefit, see Fig. 1(a).

  We also train SOTA SwinIRs of sizes 4M to 129M parameters on the same range of training set sizes, see Fig. 2(a). Albeit it performs better than the U-Net, its scaling behavior is essentially equivalent, and again after a few thousand images, only marginal benefits are

expected. Scaling up its training set and network size did, however, give benefits, the largest model we trained yields new SOTA results on four common test sets by 0.05 to 0.22dB.

- **Empirical scaling laws for compressive sensing.** We train U-Nets of sizes from 2M to 500M parameters for 4x accelerated MRI with training set sizes from 50 to 50k images from the fastMRI dataset (Knoll et al., 2020b). We again find that beyond a dataset size of about 2.5k, the rate of improvement as a function of the training set size considerably slows, see Fig. 1(c). This indicates that while models are expected to improve by increasing the training set size, we expect that training on millions of images does not significantly improves performance.

- **Empirical scaling laws for super-resolution**. We also briefly study super-resolution, and similarly as for denoising and compressive sensing find a slowing of the scaling law at moderate dataset sizes. See appendix C.

- **Understanding scaling laws for denoising theoretically.** Our empirical results indicate that the denoising performance of a neural network trained end-to-end doesn't increase as a function of training examples beyond a certain point. This is expected since once we reach a noise specific error floor, more data is not beneficial. We make this intuition precise for a linear estimator learned with early stopped gradient descent to denoise data drawn from a $d$-dimensional linear subspace. We show that the reconstruction error is upper bounded by $d/N$ plus a noise-dependent error floor level. Once the error induced by learning the signal model, $d/N$, is small relative to the error floor, more training examples $N$ are not beneficial.

Together, our empirical results show that neural networks for denoising, compressive sensing, and super-resolution applied in typical setups (i.e, Gaussian denoising with 20.17dB PSNR, and multi-coil accelerated MRI with 4x acceleration) already operate in a regime where the scaling laws for the training set size are slowing significantly and thus even very large increases of the training data are not expected to improve performance substantially. Even relatively modest model improvements such as those obtained by transformers over convolutional networks are larger than what we expect from scaling the number of training examples from tens of thousands to millions.

## 2 RELATED WORK

**Scaling laws for prediction problems.** Under the umbrella of statistical learning theory convergence rates of $1/N$ or $1/\sqrt{N}$ have been established for a range of relatively simple models and distributions, see e.g. Wainwright (2019) for an overview.

For deep neural networks used in practice, a recent line of work has empirically characterized the performance as a function of training set size and/or network size for classification and NLP (Hestness et al., 2017; Rosenfeld et al., 2019; Kaplan et al., 2020; Bahri et al., 2021; Zhai et al., 2021; Ghorbani et al., 2021; Bansal et al., 2022). In those domains, the scaling laws persist even for very large datasets, as described in more detail below. In contrast, for image reconstruction, we find that the power-law behavior already slows considerably at relatively small numbers of training examples.

Rosenfeld et al. (2019) find power-law scaling of performance with training set and network size across models and datasets for language modeling and classification. However, because the work *fixes* either training set or network size, while scaling the other, the scaling laws span only moderate ranges before saturating at a level determined by the fixed quantity and not the problem specific error floor. The papers Kaplan et al. (2020); Bahri et al. (2021); Zhai et al. (2021); Ghorbani et al. (2021); Bansal et al. (2022) including ours scale the dataset size and model size simultaneously, resulting in an improved predictive power of the obtained scaling laws.

Hestness et al. (2017) study models for language and image classification, and attribute deviations from a power-law curve to a lack of fine-tuning the hyperparameters of very large networks. For transformer language models Kaplan et al. (2020) find no deviation from a power-law for up to a training set and network size of 1B images and parameters. Further, Zhai et al. (2021) find the performance for Vision Transformers (Dosovitskiy et al., 2020) for few-shot image classification to deviate from a power-law curve only at extreme model sizes of 0.3-1B parameters and 3B images.

**The role of training set size in inverse problems.** For image reconstruction and inverse problems in general, we are not aware of work studying scaling laws in a principled manner, covering different

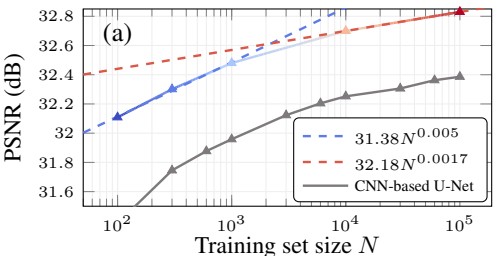 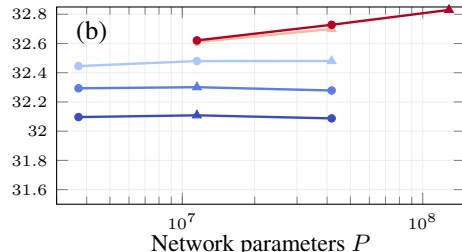

Figure 2: **Empirical scaling laws for transformer-based image denoising.** The colored curve in (a) shows the best PSNR per training set size of a SwinIR (Liang et al., 2021) with varying network sizes as depicted in (b). Colors in the plot on the left and right correspond to the same training set size. The SwinIR outperforms the U-Net (gray curve from Fig. 1(a)), but as for the U-Net the rate of improvement slows to a level that indicates that further increasing the dataset size would only marginally improve performance.

problems and model architectures. But, when proposing a new method several works study performance as a function of training set and/or network size. However, those studies typically only scale the parameter of interest while fixing all other parameters, unlike our work, which scales dataset size and network size together, which is important for identifying scaling laws. Below, we review the training set sizes, and if available their scaling properties, used by the recent SOTA in image denoising and accelerated MRI.

Zhang et al. (2017) report that for DnCNN, a standard CNN, using more than 400 distinct images with data augmentation only yields negligible improvements. Chen et al. (2021) pre-train an image processing transformer (IPT) of 115.5M network parameters on ImageNet (1.1M distinct images), and report the performance after fine-tuning to a specific task as a function of the size of the pre-training dataset. IPT's performance for denoising is surpassed by the latest SOTA in form of the CNN based DRUnet (Chen et al., 2021), the transformer based SwinIR (Liang et al., 2021) and the Swin-Conv-Unet (Zhang et al., 2022) a combination of the two. Those models have significantly fewer network parameters and were trained on a training set consisting of only ∼10k images, leaving the role of training set size in image denoising open.

The number of available training images for accelerated MRI is limited to few publicly available datasets. Hence, current SOTA (Sriram et al., 2020; Fabian & Soltanolkotabi, 2022) are trained on the largest datasets available (the fastMRI dataset), consisting of 35k images for knee and 70k for brain reconstruction (Zbontar et al., 2018; Muckley et al., 2021).

Adaptations of the MLP-Mixer (Tolstikhin et al., 2021) and the Vision Transformer (Dosovitskiy et al., 2020) to MRI (Mansour et al., 2022; Lin & Heckel, 2022) ablate their performance as a function of the training set size and find that non-CNN based approaches require more data to achieve a similar performance, but potentially also benefit more from even larger datasets. However, those experiments are run with fixed model sizes and only 3-4 realizations of training set size thus it is unclear when performance saturates.

## 3 EMPIRICAL SCALING LAWS FOR DENOISING

In this section, we consider the problem of estimating an image $\mathbf{x}$ from a noisy observation $\mathbf{y} = \mathbf{x} + \mathbf{z}$, where $\mathbf{z}$ is additive Gaussian noise $\mathbf{z} \sim \mathcal{N}(0, \sigma_z^2 \mathbf{I})$. Neural networks trained end-to-end to map a noisy observation to a clean image perform best, and outperform classical approaches like BM3D (Dabov et al., 2007) and non-local means (Buades et al., 2005) that are not based on training data.

**Datasets.** To enable studying learned denoisers over a wide range of training set sizes we work with the ImageNet dataset (Russakovsky et al., 2015). Its training set contains 1.3M images of 1000 different classes. We reserve 20 random classes for validation and testing. We design 10 training set subsets $\mathcal{S}_N$ of sizes $N \in [100, 100000]$ (see Fig. 1(a)) and with $\mathcal{S}_i \subseteq \mathcal{S}_j$ for $i \leq j$. To make the distributions of images between the subsets as homogeneous as possible, the number of images from

different classes within a subset differ by at most one. We generate noisy images by adding Gaussian noise with variance of $\sigma_z = 25$ (pixels in the range from 0 to 255), i.e. 20.17dB in PSNR.

**Model variants and training.** We train a CNN-based U-Net, a detailed description of the model architecture is in Appx. A.1. We also train the Swin-Transformer (Liu et al., 2021) based SwinIR model (Liang et al., 2021) that achieves SOTA for a variety of image reconstruction tasks, including denoising. This model is interesting since transformers scale well with the number of training examples for other domains, e.g., for image classification pre-training (Dosovitskiy et al., 2020).

For U-Net and SwinIR we vary the number of network parameters as depicted in Fig. 1 (b) and Fig. 2 (b) respectively. Appx. A.1 and A.3 contain detailed descriptions on how the models are trained and the network size is adapted.

**Results and discussion.** Fig. 1(a) and Fig. 2(a) show the reconstruction performances of the best U-Net and SwinIR respectively over all considered network sizes. Our main findings are as follows.

A training set size of around 100 is already sufficient to train a decent image denoiser. Beyond 100, we can fit a linear power law to the performance of the U-Net with a scaling coefficient $\alpha = 0.0048$ that approximately holds up to training set sizes of about 6k images. Beyond 6k, we can fit a second linear power law with significantly smaller scaling coefficient $\alpha = 0.0019$.

Similarly, for the SwinIR we can fit a power law with coefficient $\alpha = 0.0050$ in the regime of little training data and a power law with significantly smaller coefficient $\alpha = 0.0017$ in the regime of moderate training data, thus the two architectures scale essentially equivalently.

While denoising benefits from more training examples, the drop in scaling coefficient indicates that scaling from thousands to millions of images is only expected to yield relatively marginal performance gains. While the gains are small and we expect gains from further scaling to be even smaller, our largest SwinIR trained on 100k images yields a new SOTA for Gaussian color image denoising. On four common test sets it achieves an improvement between 0.05 to 0.22dB, see Appx. B. Visualizations of reconstructions from models along the curves in Fig. 1(a) and Fig. 2(a) can be found in Fig. 4, Appx. A.

Comparing the U-Net scaling to SwinIR scaling (Fig. 2(a)) we see that the effect of large training sets on the performance of the U-Net can not make up for the improved modeling error that stems from the superior network architecture of the SwinIR. In fact, the simulated scaling coefficient $\alpha = 0.0019$ predicts a required training set size of about 150M images for U-Net to achieve the best performance of the SwinIR, and that is an optimistic prediction since it assumes that the scaling does not slow down further for another 3 orders of magnitude. Thus, model improvements such as those obtained by transformers over convolutional networks are larger than what we expect from scaling the number of training examples from tens of thousands to millions.

An interesting question is whether different noise levels lead to different data requirements, as indicated by a different scaling behavior. We investigate the performance of U-Net for a smaller noise variance $\sigma_z = 15$ in Appx. D.2. While the results are still preliminary, the qualitative scaling behavior is similar; the difference are that the scaling law pertaining to smaller noise is slightly steeper, and overall the performance improves by about 1.3dB in PSNR.

Finally, note that for our results we choose to re-sample the noise per training image in every training epoch, since it makes the best use of the available clean images. However, fixing the noise is also interesting, since it is more similar to a setup in which the noise statistics are unknown like in real-world noise removal. See Appx. D.1 for additional results where the noise is fixed. We found that compared to re-sampling the noise the performance of a U-Net drops by 0.3dB for small training set sizes. As the training set size increases the performance approaches the one of re-sampling the noise resulting in a steeper scaling law that flattens at slightly larger training set sizes.

**Robustness of our finding of slowing scaling laws.** Our main finding for denoising is that the scaling laws slow at relatively small amounts of training data points (i.e., the scaling coefficient becomes very small). We next argue why we think that this is a robust finding.

In principle it could be that the networks are not sufficiently large for the performance to improve further as the dataset size increases. However, for the U-Net and training set sizes of 10k and 30k the

performance increase already slows, and for those training set sizes, increasing the network size does not improve performance, as shown in Fig. 1(b). For smaller network sizes the curves in Fig. 1(b) first increase and then decrease indicating that we found network sizes close to the optimum.

The parameter scaling of the SwinIR in Fig. 2(b) indicates that for training set sizes of 10k and 100k the performance still benefits from larger networks. Hence, the power law for those training set sizes in Fig. 2(a) can become slightly steeper once we further increase the network size. However, for the slope of the flat power law in (a) to reach the slope of the steep power law, the parameter scaling in (b) for 100k training images would need to hold for another 2 orders of magnitude, i.e. about 12B instead of the currently used 129M network parameters, which is very unlikely considering that the current network sizes already suffice to saturate the performance for small/moderate training set sizes.

Finally, it could be that with higher quality or higher diversity of the data, the denoising performance would further improve. In our experiments, we increase the training set sizes by adding more images from the same classes of ImageNet, and we continue to test on different classes. Thus, it could be that adding training examples from a different, more diverse data source leads to less of a slowing of the scaling law in Fig. 1(a). We test this hypothesis by scaling our training set with 3k images to 10k images not by adding more images from ImageNet but images from the datasets that are used to train the original SwinIR, i.e., we add all images from DIV2K (Agustsson & Timofte, 2017), Flickr2K (Timofte et al., 2017) and BSD500 (Arbeláez et al., 2011) and add the remaining images from WED (Ma et al., 2017). Keeping all other hyperparameters fixed the U-Net obtains for this dataset a PSNR of 32.239dB, which is slightly worse than the 32.252dB it achieves with our training set with 10k images only from ImageNet. Hence, we reject the hypothesis that the drop in scaling coefficients can be explained with how the training sets are designed in our experiment.

## 4 EMPIRICAL SCALING LAWS FOR COMPRESSIVE SENSING

We consider compressive sensing (CS) to achieve 4x accelerated MRI. MRI is an important medical imaging technique for its non-invasiveness and high accuracy. Due to the physics of the measurement process, MRI is inherently slow. Accelerated MRI aims at significantly reducing the scan time by undersampling the measurements. In addition, MRI scanners rely on parallel imaging, which uses multiple receiver coils to simultaneously collect different measurements of the same object.

This leads to the following reconstruction problem. We are given measurements $\mathbf{y}_i \in \mathbb{C}^m$ of the image $\mathbf{x} \in \mathbb{C}^n$ as $\mathbf{y}_i = \mathbf{MFS}_i\mathbf{x} + \text{noise}_i$ for $i = 1, ..., C$, and our goal is to reconstruct the image from those measurements. Here, $C$ is the number of receiver coils, the diagonal matrices $\mathbf{S}_i \in \mathbb{C}^{n \times n}$ are the sensitivity maps modelling the signal strength perceived by the $i$-th coil, $\mathbf{F} \in \mathbb{C}^{n \times n}$ is the discrete Fourier transform (DFT), and $\mathbf{M} \in \mathbb{C}^{m \times n}$ is a binary mask implementing the undersampling.

Classical CS approaches (Lustig et al., 2008) first estimate the sensitivity maps with a method such as ESPIRiT (Uecker et al., 2014), and second estimate the unknown image by solving a regularized optimization problem, such as total-variation norm minimization. Recently, deep learning based methods have been shown to significantly outperform classical CS methods due to their ability to learn more complex and accurate signal models from data (Knoll et al., 2020a; Muckley et al., 2021).

**Datasets.** To explore the performance of learning based MR reconstruction over a wide range of training set sizes, we use the fastMRI multi-coil brain dataset (Knoll et al., 2020b), which is the largest publicly available dataset for MRI. The dataset consists of images of different contrasts. To ensure that the statistics of the dataset are as homogeneous as possible, we take the subset of the dataset corresponding to a single contrast resulting in 50k training images. We design training set subsets $\mathcal{S}_N$ of size $N \in [50, 50000]$ (see Fig. 1 (c)) with $\mathcal{S}_i \subseteq \mathcal{S}_j$ for $i \leq j$. For more information on selection, division and subsampling of the dataset see Appx. A.2.

**Model variants and training.** We train the same U-Net model used in Section 3 and described in Appx. A.1 but with 4 blocks per encoder/decoder. The network is trained end-to-end to map a coarse reconstruction $\hat{\mathbf{x}}_{\text{CR}}$ to the ground truth image $\mathbf{x}$. The coarse reconstruction is obtained as $\hat{\mathbf{x}}_{\text{CR}} = \left( \sum_{i=1}^{C} |\mathbf{F}^{-1}\mathbf{y}_{i,\text{ZF}}|^2 \right)^{1/2}$, where $\mathbf{F}^{-1}$ is the inverse DFT, and $\mathbf{y}_{i,\text{ZF}}$ are the undersampled measurements of the $i$-th coil, where missing entries are filled with zeros. We vary the number of

network parameters as depicted in Fig. 1 (d). Appx. A.2 contains detailed descriptions on how the model is trained and the network size is adapted.

**Results and discussion.** For each training set size Fig. 1 (c) shows the reconstruction performance in structural similarity (SSIM) of the best model over all simulated network sizes. There are 2 main findings. First, we can fit a linear power law with a scaling coefficient $\alpha = 0.0072$ that holds up to training set sizes of about 2.5k images. Second, for training set sizes starting from 5k we can fit a second linear power law but with significantly smaller scaling coefficient $\alpha = 0.0026$.

Similar to denoising in Section 3 we conclude that while accelerated MRI slightly benefits form more training examples, the drop in scaling coefficient indicates that it is unlikely that even training set sizes of the order of hundreds of millions of images result in substantial gains. Visualizations of reconstructions from models along the curve in Fig. 1 (c) can be found in Fig. 5, Appx. A.2.

**Robustness of our finding of slowing scaling laws.** Next, we argue why the drop in scaling coefficient for accelerated MRI is a robust finding.

In Fig. 1 (d) we demonstrate that the network size does not bottleneck the performance of our largest training sets. For each training set size the performance as a function of the number of network parameters is relatively flat. Even small training sets benefit from large networks before their performance slightly decays for very large networks. Hence, we expect that for large training sets the performance as a function of network parameters would not increase significantly before decaying.

Finally, is there a different type of training data that would improve the scaling coefficient? We don't think so, since all experiments are for the very specific task of reconstructing brain images of one particular contrast, which means that the examples that we add to increase the size of the training sets is already the data most suitable to solve this task.

## 5 UNDERSTANDING SCALING LAWS FOR DENOISING THEORETICALLY

We study a simple linear denoiser that is trained end-to-end to reconstruct a clean image from a noisy observation, in order to understand how the error as a function of the number of training examples for inverse problems is expected to look like.

We define a joint distribution over a signal and the corresponding measurement $(\mathbf{x}, \mathbf{y})$ as follows. Consider a $d < n$ dimensional subspace of $\mathbb{R}^n$ parameterized by the orthonormal basis $\mathbf{U} \in \mathbb{R}^{n \times d}$. We draw a signal approximately uniformly from the subspace $\mathbf{x} = \mathbf{U}\mathbf{c}$, where $\mathbf{c} \sim \mathcal{N}(0, \mathbf{I})$, and an associated noisy measurement as $\mathbf{y} = \mathbf{x} + \mathbf{z}$, where $\mathbf{z} \sim \mathcal{N}(0, \sigma_z^2 \mathbf{I})$ is Gaussian noise. The subspace is unknown, but we are given a training set $\{(\mathbf{x}_1, \mathbf{y}_1), \ldots, (\mathbf{x}_N, \mathbf{y}_N)\}$, consisting of examples drawn iid from the joint distribution over $(\mathbf{x}, \mathbf{y})$.

We assume that the signal lies in a low dimensional subspace. Assuming that data lies in a low-dimensional subspace or more general, a union of low-dimensional subspaces, is common, for example it underlies the denoising of natural images via wavelet thresholding (Donoho & Johnstone, 1995; Simoncelli & Adelson, 1996; Chang et al., 2000). Mohan et al. (2020) found that even deep learning based denoisers implicitly perform a projection onto an adaptively-selected low-dimensional subspace that captures the features of natural images.

We consider a linear estimator of the form $f_{\mathbf{W}}(\mathbf{y}) = \mathbf{W}\mathbf{y}$, and measure performance in terms of the expected mean-squared reconstruction error, defined as $R(\mathbf{W}) = \frac{1}{d}\mathbb{E}\left[\|\mathbf{W}\mathbf{y} - \mathbf{x}\|_2^2\right]$, where expectation is over the joint distribution of $(\mathbf{x}, \mathbf{y})$.

**The optimal linear estimator.** The optimal linear estimator (i.e., the estimator that minimizes the risk $R$) is given by $\mathbf{W}^* = \frac{1}{1+\sigma_z^2}\mathbf{U}\mathbf{U}^T$. The estimator projects the data onto the subspace and shrinks towards zero, depending on the noise variance. The associated risk is $R(\mathbf{W}^*) = \sigma_z^2/(1 + \sigma_z^2)$.

**An estimator based on subspace estimation.** The optimal estimator $\mathbf{W}^*$ requires knowledge of the unknown subspace. We can learn the subspace from the noisy data by performing principal component analysis on $\mathbf{Y} = [\mathbf{y}_1, \ldots, \mathbf{y}_N]$. Specifically, we estimate the subspace as the $d$-leading

singular vectors of the empirical co-variance matrix $\mathbf{Y}\mathbf{Y}^T \in \mathbb{R}^{n \times n}$, denoted by by $\hat{\mathbf{U}} \in \mathbb{R}^{n \times d}$. If the measurement noise is zero (i.e., $\sigma_z^2 = 0$), $\hat{\mathbf{U}}$ is an orthonormal basis for the $d$-dimensional subspace $\mathbf{U}$ provided we observe at least $d$ many linearly independent signals, which occurs with probability one if we draw data according to the model defined above, and if the number of training examples obeys $N \geq d$. There is a vast literature on PCA; see Vershynin (2011; 2018); Rudelson & Vershynin (2010); Tropp (2012) for tools from high-dimensional probability to analyze the PCA estimate.

Now consider the estimator $\mathbf{W}_{\text{PCA}} = \frac{1}{1+\sigma_z^2}\hat{\mathbf{U}}\hat{\mathbf{U}}^T\mathbf{y}$ based on $N$ noisy training points. The estimator assumes knowledge of the noise variance, but it is not difficult to estimate it relatively accurately. The following result, proven in Appx. F, characterizes the associated risk.

**Theorem 1.** *Suppose that the number of training examples obeys $(d + n\sigma_z^2)\log(n) \leq N$. For a numerical constant $c$, with probability at least $1 - n^{-10} - 3e^{-d} + e^{-n}$, the risk of the PCA-estimate is bounded by*

$$R(\mathbf{W}_{PCA}) \leq R(\mathbf{W}^*) + c(d + n\sigma_z^2)\log(n)/N.$$

Thus, as long as the number of training examples $N$ is sufficiently large relative to $(d + n\sigma_z^2)$, the risk of the PCA-estimator is close to the risk of the optimal estimator.

**Estimator learned end-to-end.** We now consider an estimator learned end-to-end, by applying gradient descent to the empirical risk $\mathcal{L}(\mathbf{W}) = \sum_{i=1}^{N} \|\mathbf{W}\mathbf{y}_i - \mathbf{x}_i\|_2^2$ and we regularize via early-stopping the gradient descent iterations. The risk of the estimate after $k$ iterations of gradient descent, $\mathbf{W}^k$, is bounded by the next result. This estimator mimics the supervised training we consider throughout this section, with the difference that here we consider a simple neural network, and in the previous sections we trained a neural network.

**Theorem 2.** *Let $\mathbf{W}^k$ be the matrix obtained by applying $k$ iterations of gradient descent with stepsize $\eta$ starting at $\mathbf{W}^0 = 0$ to the loss $\mathcal{L}(\mathbf{W})$. Consider the regime where the number of training examples obeys $(d + n\sigma_z^2)\log(n) \leq N \leq \xi d/\sigma_z^2$ for an arbitrary $\xi$ and $N\log(N) \leq n$. For an appropriate choice of the stepsize $\eta$ there exists an optimal early stopping time $k_{opt}$ at which the risk of the estimator $f_{\mathbf{W}}(\mathbf{y}) = \mathbf{W}^{k_{opt}}\mathbf{y}$ is upper-bounded with probability at least $1 - 2e^{-N/8} - 2e^{-N/18} - 5n^{-9} - 5e^{-d} - 2e^{-n} - e^{-N} - 2e^{-n/2}$ by*

$$R(\mathbf{W}^{k_{opt}}) \leq (8 + 2\xi)R(\mathbf{W}^*) + c\left(1 + \frac{n\sigma_z^2}{d}\right)\log n \frac{(d + n\sigma_z^2)\log(n)}{N} + c\xi\sqrt{\frac{(d + \sigma_z^2 n)\log n}{N}},$$

*where $c$ is a numerical constant.*

The proof is provided in Appx. G. Similar to Thm. 1 the first term in the bound corresponds to the noise-dependent error floor and the second two terms decrease in $(d + n\sigma_z^2)/N$ and represent the error induced by learning the estimator with a finite number of training examples. Once this error is small relative to the error floor, more training examples do not improve performance. See Appx. E for a more detailed discussion on the estimator and the assumptions of Thm. 2.

**Discussion.** In Fig. 3, we plot the risk as a function of the training set size for the early-stopped empirical risk minimization (ERM) and the PCA based estimator. Starting at a training set size of $N = 100$, we see that the risk minus the optimal risk follows approximately a power law, i.e., $\log(R(\mathbf{W}) - R(\mathbf{W}^*)) \approx -\alpha \log(N)$, as suggested by Thms. 1 and 2.

In practice, however, we don't know the risk of the optimal estimator, $R(\mathbf{W}^*)$. Therefore, in the second row of Fig. 3 and throughout our empirical results we plot the risk $\log(R(\mathbf{W}))$ as a function of the number of training examples $\log(N)$ and distinguish different regions by fitting approximated scaling coefficients $\alpha$ to $\log(R(\mathbf{W})) \approx -\alpha \log(N)$. In our theoretical example, we can identify three regions, coined in Hestness et al. (2017): The *small data region* in which the training set size does not suffice to learn a well-performing mapping and the *power-law region* in which the performance decays approximately as $N^\alpha$ with $\alpha < 0$, and an problem-specific *irreducible error region*, which is $R(\mathbf{W}^*)$ here. Note that the power-law coefficient in the risk plot $R(\mathbf{W})$ are smaller then those when plotting $R(\mathbf{W}) - R(\mathbf{W}^*)$, an are only approximate scaling coefficients.

Already in this highly simplified setup of an inverse problem the true scaling coefficients heavily depend on model parameters such as the noise variance as can be seen in Fig. 3. The scaling coefficients further vary with the signal dimension $d$ and ambient dimension $n$ (see Appx. E.1).

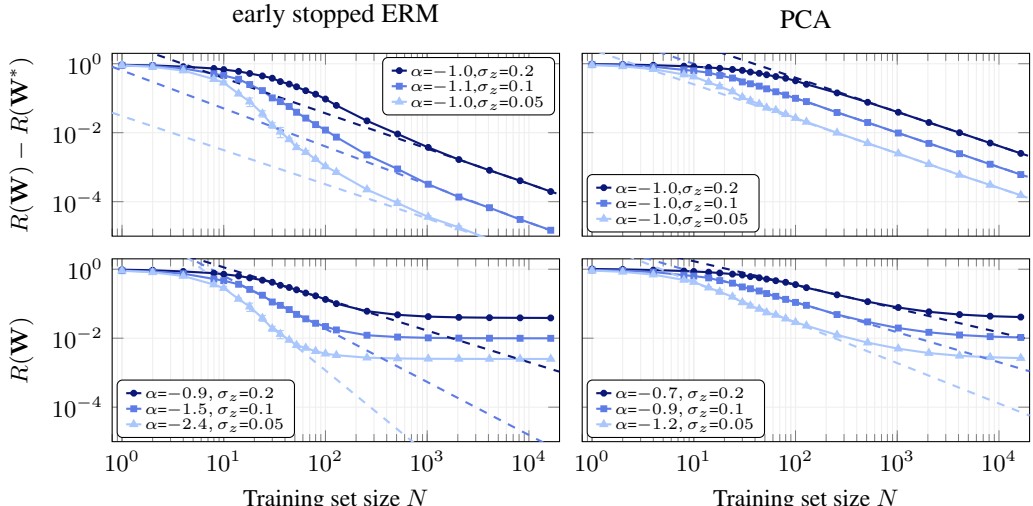

Figure 3: **Subspace denoising.** Simulated risks of the early stopped ERM estimator $\mathbf{W}^{k_{opt}}$ (left) and the PCA estimator $\mathbf{W}_{\mathrm{PCA}}$ (right) over the training set size $N$. The signal and ambient dimensions are $d = 10, n = 1000$ and we vary the noise level $\sigma_z$. We fit linear scaling laws $\sim N^{\alpha}$ to the power law regions. As the noise level decreases the scaling coefficients $\alpha$ increase. Further, the learned estimator exhibits steeper scaling than the PCA estimator. Error bars are over 5 independent runs.

## 6 CONCLUSION

We found that the performance improvement of deep learning based image reconstruction as a function of the number of training examples slows already at moderate training set sizes, indicating that only marginal gains are expected beyond a few thousand examples.

**Limitations.** This finding is based on studying three different reconstruction problems (denoising, super-resolution, and compressive sensing), two architectures (U-net and SwinIR), and for each setup we extensively optimized hyperparameters and carefully scaled the networks. Scaling gave new state-of-the-art results on four denoising datasets, which provides some confidence in the setup.

Moreover, our statements necessarily pertain to the architectures and metrics we study. It is possible that other architectures or another scaling of architectures can yield larger improvements when scaling architectures. It is also widely acknowledged that image quality metrics (such as SSIM and PSNR) do not fully capture the perceived image quality by humans, and it could be that more training data yields improvements in image quality that are not captured well by SSIM and PSNR.

Most importantly, our findings pertain to standard in-distribution evaluation, i.e., the test images are from the same distribution as the training images. To achieve robustness against testing on images out of the training distribution recent work indicates a positive effect of larger and more diverse training sets (Miller et al., 2021; Darestani et al., 2022; Nguyen et al., 2022). Thus it is possible that the out-of-distribution performance of image reconstruction methods improves when trained on a larger and a more diverse dataset, even though the in-distribution performance does not improve further.

**Future research.** We focus on supervised image reconstruction methods. For image denoising and other image reconstruction problems self-supervised approaches are also performing very well (Laine et al., 2019; Wang et al., 2022; Zhou et al., 2022), even though supervised methods perform best if clean images are available. It would be very interesting to investigate the scaling laws for self-supervised methods; perhaps more training examples are required to achieve the performance of a supervised setup. We leave this for future work.

REPRODUCIBILITY

The repository at `https://github.com/MLI-lab/Scaling_Laws_For_Deep_Learning_Based_Image_Reconstruction` contains the code to reproduce all results in the main body of this paper.

ACKNOWLEDGMENTS

The authors acknowledge support by the Institute of Advanced Studies at the Technical University of Munich, the Deutsche Forschungsgemeinschaft (DFG, German Research Foundation) - 456465471, 464123524, the DAAD, and the German Federal Ministry of Education and Research and the Bavarian State Ministry for Science and the Arts. The authors of this work take full responsibility for its content.

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

# A    DETAILS OF THE EXPERIMENTAL SETUPS

## A.1    EXPERIMENTAL DETAILS FOR EMPIRICAL SCALING LAWS FOR DENOISING WITH A U-NET

In this Section, we give a detailed description of the experimental setup that led to our results for Gaussian denoising with a U-Net presented in Fig.1 (a),(b) and Section 3.

In addition, Fig. 4 shows examples of reconstructions from different models along the performance curve in Fig. 1 (a). In these examples the improvement in perceived image quality from increasing the training set size from 100 to 1000 is larger than from increasing from 1000 to 10000 or from 10000 to 100000. This correlates with our quantitative findings in Fig. 1 (a).

Next, we describe the experimental details. We train U-Nets with two blocks in the encoder and decoder part respectively and skip connections between blocks. Each block consists of two convolutional layers with LeakyReLU activation and instance normalization (Ulyanov et al., 2017) after every layer, where the number of channels is doubled (halved) after every block in the encoder (decoder). The downsampling in the encoder is implemented as average pooling and the upsampling in the decoder as transposed convolutions. As proposed by Zhang et al. (2017) we train a residual denoiser that learns to predict $\mathbf{y} - \mathbf{x}$ instead of directly predicting $\mathbf{x}$, which improves performance.

We scale the network size by increasing the width of the network by scaling the number of channels per (transposed) convolutional layer. For denoising we trained U-Nets of 7 different sizes. We vary the number of channels in the first layer in $\{16, 32, 64, 128, 192, 256, 320\}$, which corresponds to $\{0.1, 0.5, 1.9, 7.4, 16.7, 30.0, 46.5\}$ million parameters. We do not vary the depth, since this would change the dimension of the informational bottleneck in the U-Net and thus change the model family itself.

The exact training set sizes we consider are $\{0.1, 0.3, 0.6, 1, 3, 6, 10, 30, 60, 100\}$ thousand images from ImageNet. We center crop the training images to $256 \times 256$ pixels. We also tried a smaller patch size of $128 \times 128$ pixels, but larger patches showed to have a better performance in the regime of large training set sizes, which is why the results presented in the main body are for patch size $256 \times 256$. For a comparison between the scaling behavior of the two different patch sizes see Fig. 9, Appx. D.3.

We do not use any data augmentation, since it is unclear how to account for it in the number of training examples. For validation and testing we use 80 and 300 images respectively taken from 20 random classes that are not used for training.

We use mean-squared-error loss and Adam (Kingma & Ba, 2014) optimizer with $\beta_1 = 0.9, \beta_2 = 0.999$. For moderate training set sizes up to 3000 images we find that heuristically adjusting the initial learning rate with the help of an automated learning rate annealing performs well. To this end, we start with a learning rate of $10^{-4}$ and increase after every epoch by a factor of 2 until the validation loss does not improve for 3 consecutive epochs. We then load the model checkpoint from the learning rate that was still performing well and continue with that learning rate. We observe that this scheme typically picks an initial learning rate reduced by factor 2 for every increase in the number of channels $\mathcal{C}$. For training set sizes larger than 3000 we directly apply this rule to pick an initial learning rate without annealing as we found this to give slightly better results. In particular, for number of channels $\mathcal{C} = \{128, 192, 256, 320\}$ we used initial learning rates $\eta = \{0.0032, 0.0016, 0.0008, 0.0004\}$.

For small training sets up to 1000 images we found that batch size of 1 works best. For larger sets we use a batch size of 10 and found that further increasing the batch size does not improve performance.

We do not put a limit on the amount of compute for training. We use an automated learning rate decay that reduces the learning rate by $0.5$ if the validation PSNR has not improved by at least 0.001 for 10 epochs or 6 epochs for training set sizes starting from 6000 images. Once the learning rate drops to $10^{-5}$ we observe near to no gains in validation loss and stop the training after 10 additional epochs.

For training set sizes up to 1000 we train 3 random seeds and pick the best run. As the variance between runs compared to the gain in performance between different training set sizes decreases with increasing training set size we only run one seed for larger training set sizes.

The experiments were conducted on four NVIDIA A40, four NVIDIA RTX A6000 and four NVIDIA Quadro RTX 6000 GPUs. We measure the time in GPU hours until the best epoch according to the validation loss resulting in about 1800 GPU hours for the experiments in Fig. 1(a),(b).

## A.2 EXPERIMENTAL DETAILS FOR EMPIRICAL SCALING LAWS FOR COMPRESSIVE SENSING WITH A U-NET

In this Section, we give a detailed description of the experimental setup that led to our results for compressive sensing MRI presented in Fig.1 (c),(d) and Section 4.

In addition, Fig. 5 shows examples of reconstructions from different models along the performance curve in Fig. 1 (c). In these examples the improvement in perceived image quality from increasing the training set size from 500 to 2500 is larger than from increasing from 2500 to 10000 or from 10000 to 50000. This correlates with our quantitative findings in Fig. 1 (c).

The first in row in Fig. 5 shows an example in which all models including the one trained on the largest training set fail to recover a fine detail. This is a known problem in accelerated MRI and has been documented in both editions of the fast MRI challenge (Knoll et al., 2020a; Muckley et al., 2021) in which for all methods examples could be found in which fine details have not been recovered. However, the question remains if more training data or better models would help or the details are simply not there since the information is lost due to the large undersampling factors considered in the fastMRI challenges and in this work.

Next, we describe the experimental details. For compressed sensing in the context of accelerated MRI we trained U-Nets of 14 different sizes. We vary the number of channels in the first layer in $\{16, 32, 48, 64, 96, 112, 128, 144, 160, 176, 192, 208, 224, 256\}$, which corresponds to $\{2, 8, 18, 31, 70, 95, 124, 157, 193, 234, 279, 327, 380, 496\}$ million network parameters.

The exact training set sizes we consider are $\{0.05, 0.25, 0.5, 1, 2.5, 5, 10, 25, 50\}$ thousand AXT2 weighted images from fastMRI multi-coil brain dataset (Zbontar et al., 2018), where AXT2 corresponds to all images of one type of contrast. We focused on images only from this type to make the statistics of our datasets as homogeneous as possible. We do not use any data augmentation, since it is unclear how to account for it in the number of training examples. We use 4732 and 730 additional images for testing and validation.

We consider an acceleration factor of 4 meaning that we only measure 25% of the information. We obtain the 4 times undersampled measurements by masking the fully sampled measurement $\mathbf{y}$ with an equispaced mask with 8% center fractions meaning that in the center of $\mathbf{y}$ we take all the measurements and take the remaining ones at equispaced intervals.

We use structural similarity (SSIM) loss and RMSprop optimizer with $\alpha = 0.99$ as this is the default in the fastMRI repository (Zbontar et al., 2018) and we found no improvement by replacing it with Adam. We do not put a limit on the amount of compute invested into training. We deploy an automated learning rate decay that starts at a learning rate of $10^{-3}$ and decays by a factor 0.1 if the validation SSIM has not improved by at least $10^{-4}$ for 5 epochs. Once the learning rate drops to $10^{-6}$ we stop the training after 10 additional epochs. Only for the largest training set sizes 25k,50k we found that an additional drop to $10^{-7}$ resulted in further performance gains. We use a batch size of 1.

For training set sizes up to 5k we train three models with random seeds and pick the best. For larger training set sizes we only run one seed, since the variance between runs decreased.

The experiments were conducted on four NVIDIA A40, four NVIDIA RTX A6000 and four NVIDIA Quadro RTX 6000 GPUs. We measure the time in GPU hours until the best epoch according to the validation loss resulting in about 4250 GPU hours for the experiments in Fig. 1 (c),(d).

## A.3 EXPERIMENTAL DETAILS FOR EMPIRICAL SCALING LAWS FOR DENOISING WITH THE SWINIR

In this Section, we give a detailed description of the experimental setup that led to our results for Gaussian denoising with a SwinIR presented in Fig.2 (a),(b) and Section 3.

In addition, Fig. 4 shows examples of reconstructions from different models along the performance curve in Fig. 2 (a). Despite of the best SwinIR clearly outperforming the best U-Net in terms of PSNR

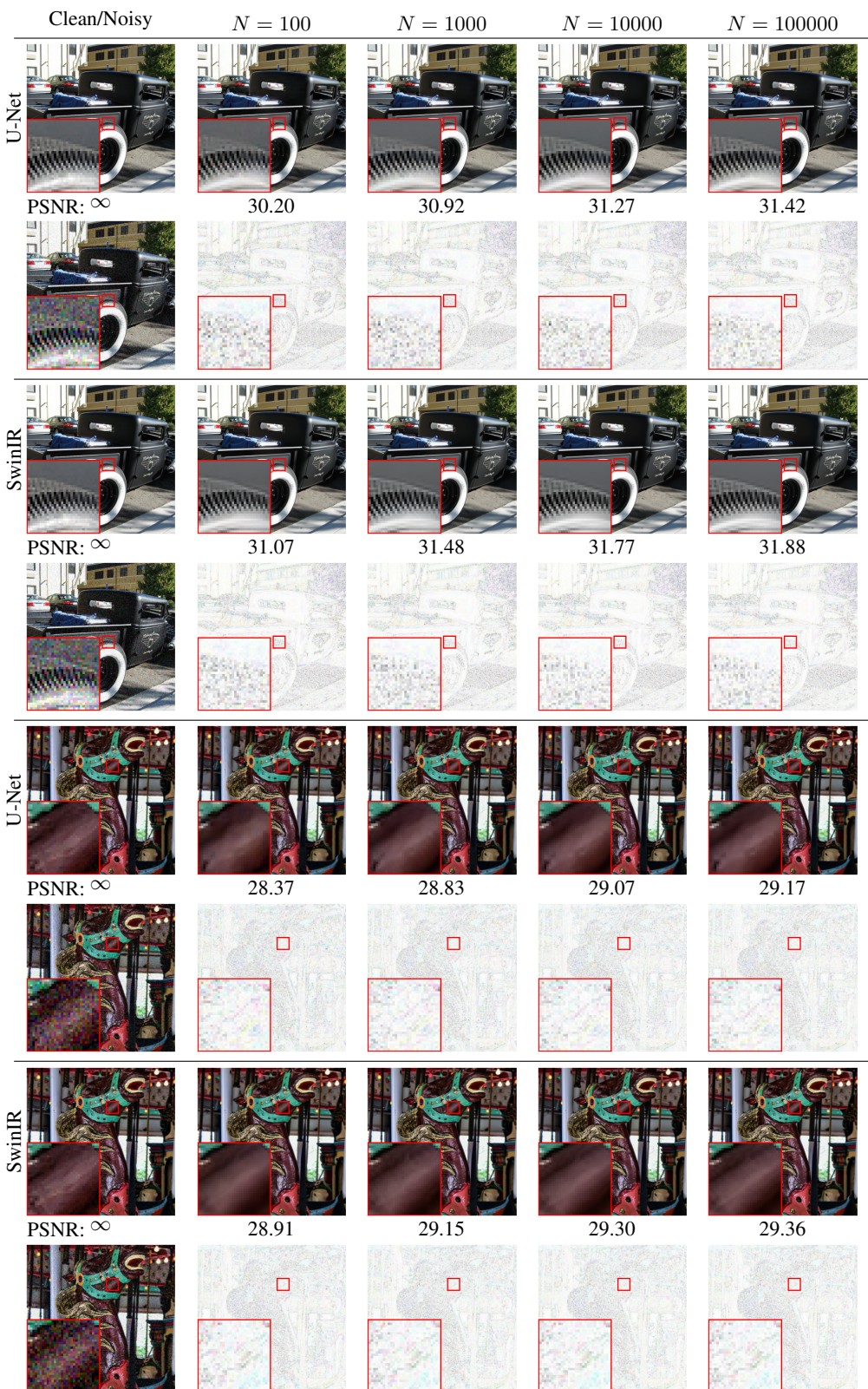

Figure 4: **Reconstructions along the scaling law for denoising with a U-Net and SwinIR.** The two examples illustrate how the reconstruction quality improves as the training set size $N$ increases. First rows: ground truth and reconstruction, Second rows: residuals w.r.t. the ground truth.

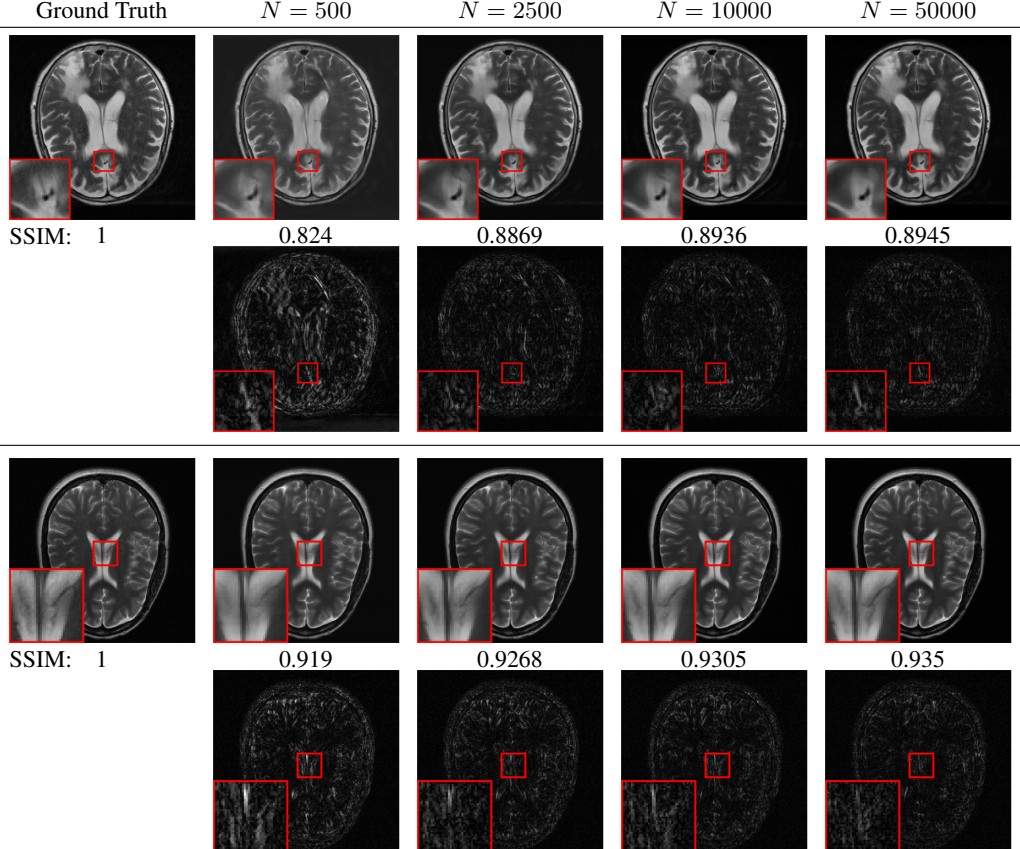

Figure 5: **Reconstructions along the scaling law for compressive sensing MRI with a U-Net.** The two examples illustrate how the reconstruction quality improves as the training set size $N$ increases. For each example the first row shows the ground truth and the reconstruction with the respective quantitative performance in SSIM and the second row shows the residual of each reconstruction with respect to the ground truth.

Table 1: **Batch size and maximal number of steps for every experiment in Fig. 2.** Each experiment can be identified by the number of training examples $N$ in thousands and the network size small(S)/middle(M)/large(L)/huge(H).

| $N/P$ | 0.1/S | 0.1/M | 0.1/L | 0.3/S | 0.3/M | 0.3/L | 1.0/S |
|---|---|---|---|---|---|---|---|
| Batch size | 8 | 20 | 6 | 8 | 20 | 6 | 8 |
| # epochs $(\cdot 10^3)$ | 11.52 | 11.52 | 11.52 | 5.76 | 5.76 | 5.76 | 3.84 |
| $N/P$ | 1.0/M | 1.0/L | 10/M | 10/L | 100/M | 100/L | 100/H |
| Batch size | 20 | 6 | 20 | 8 | 20 | 8 | 4 |
| # epochs $(\cdot 10^3)$ | 4.32 | 3.855 | 1.232 | 1.28 | 0.128 | 0.128 | 0.128 |

Table 2: **List of used network configurations of the SwinIR Liang et al. (2021).** We consider four network sizes small(S)/middle(M)/large(L)/huge(H) by varying the number of residual Swin Transformer blocks, the number of Swin Transformers per block, the number of attention heads per Swin Transformer, the number of channels in the input embedding and the width of the fully connected layers in a Swin Transformer.

| Size | # blocks | # transformers | # heads | # channels | MLP width | learning rate |
|---|---|---|---|---|---|---|
| S | 5 | 5 | 6 | 120 | 240 | $2 \cdot 10^{-4}$ |
| M | 6 | 6 | 6 | 180 | 360 | $2 \cdot 10^{-4}$ |
| L | 8 | 8 | 6 | 240 | 720 | $1 \cdot 10^{-4}$ |
| H | 11 | 8 | 8 | 360 | 720 | $5 \cdot 10^{-5}$ |

it is difficult for the naked eye to notice large differences in the quality of the reconstructions. This indicates that for Gaussian denoising both models already operate in a regime, where improvements to be made are only marginal.

Next, we describe the experimental details. To obtain Fig. 2 we train the SwinIR for color denoising from Liang et al. (2021) on the same training sets mentioned in Appx. A.1 with $\{0.1, 0.3, 1, 10, 100\}$ thousand images from ImageNet. Instead of center cropping the training images to $256 \times 256$ we have to crop to $128 \times 128$ pixels as larger input patches would make it computational infeasible for us to train large versions of the SwinIR. The largest SwinIR alone took over 2 months to train on 4 NVIDIA A40 GPUs.

We train 4 different network sizes with $\{3.7, 11.5, 41.8, 128.9\}$ million parameters. We denote the four network sizes as small(S)/middle(M)/large(L)/huge(H).

The training details and network configurations are as follows. The default SwinIR for denoising (Liang et al., 2021) was proposed for a training set size of about 10k images, 11.5M network parameters and was trained with a batch size of 8 for $T = 1280$ epochs, where the learning rate is halved at $[0.5T, 0.75T, 0.875T, 0.9375T]$ epochs. We keep the learning rate schedule but adjust the maximal number of epochs $T$ according to the training set size. Table 1 shows batch size and maximal number of epochs for every experiment in Fig. 2. We did not optimize over the choice of the batch size but picked the batch size as prescribed by the availability of computational resources.

See Liang et al. (2021) for a detailed description of the SwinIR network architecture. We vary the network size by adjusting the number of residual Swin Transformer blocks, the number of Swin Transformers per block, the number of attention heads per Swin Transformer, the number of channels in the input embedding and the width of the fully connected layers in a Swin Transformer. Table 2 contains a summary of the settings. When scaling up the network size, we invested in the parameters that seemed to be most promising in the ablation studies in Liang et al. (2021).

The experiments were conducted on four NVIDIA A40 and four NVIDIA RTX A6000. We used about 13000 GPU hours for the tranformer experiments in Fig. 2. For training the models in parallel on multiple GPUs we utilize the `torch.distributed` package with the `glow` backend, instead of the faster `nccl` backend, which was unfortunately not available on our hardware at that time.

Table 3: **Benchmarking results for Gaussian color image denoisng.** Average PSNR on 4 common test sets of our best U-Net (46.5M parameters, 100k training images) and three different versions of the SwinIR (see Appx. A.3). Values for the original SwinIR and SCUnet are taken from Liang et al. (2021) and Zhang et al. (2022). Best and second best performance are in red and blue colors respectively.

| Dataset | Noise level | U-Net | SwinIR original | SCUnet original | SwinIR 10/M | SwinIR 10/L | SwinIR 100/H |
|---------|-------------|-------|-----------------|-----------------|-------------|-------------|--------------|
| CBSD68 | 25 | 31.57 | 31.78 | 31.79 | 31.72 | 31.78 | 31.84 |
| Kodak24 | 25 | 32.78 | 32.89 | 32.92 | 32.97 | 33.05 | 33.14 |
| McMaster | 25 | 33.01 | 33.20 | 33.34 | 33.20 | 33.32 | 33.44 |
| Urban100 | 25 | 32.17 | 32.90 | 33.03 | 32.63 | 32.90 | 33.21 |

## B  BENCHMARKING OUR MODELS FOR GAUSSIAN IMAGE DENOISING

In this Section, we evaluate the models we trained for image denoising in Section 3 on four common test sets form the literature and show that the largest SwinIR trained on the largest dataset achieves new SOTA for all four test sets and the considered noise level.

In Fig. 2 in the main body we compared the performance of the U-Nets and the SwinIRs trained on subsets of ImageNet for image denoising. The models are evaluated on a test set sampled from ImageNet. We observed that while SwinIRs significantly outperform U-Nets, the performance gain from increasing the training set size slows already at moderate training set sizes for both architectures equally. However, there is still a moderat performance gain in scaling the models, and thus we expect the largest SwinIR trained on the largest dataset to outperform the original SwinIR from Liang et al. (2021). Our results in this section show that this is indeed the case.

In Table 3 we evaluate on the standard test sets for Gaussian color image denoising CBSD68 (Martin et al., 2001), Kodak24 (Franzen, 1999), McMaster (Zhang et al., 2011) and Urban100 (Huang et al., 2015). We observe a significant performance difference between the best U-Net (46.5M parameters, 100k training images) and the other transformer based methods. As expected, our largest SwinIR trained on the largest dataset SwinIR 100/H outperforms the original SwinIR, but also the SCUnet (Zhang et al., 2022) a later SOTA model that has been demonstrated to outperform the original SwinIR.

We also depict the gains for the SwinIR from just scaling up the network size and then from scaling up network size and training set size. While this led to a new SOTA for Gaussian image denoising, note that on the downside training the SwinIR 10/M, which is comparable to the original SwinIR, took about 2 weeks on 4 NVIDIA A40 gpus, while training the SwinIR 100/H took over 2 months.

## C  EMPIRICAL SCALING LAWS FOR IMAGE SUPER-RESOLUTION WITH A U-NET

In this Section, we consider the problem of super-resolution, i.e., estimating an high-resolution image from a low-resolution version of the image. This can be viewed as a compressive sensing problem, since we can view the super-resolution problem as reconstructing a signal $\mathbf{x} \in \mathbb{R}^n$ from a downsampled version $\mathbf{y} = \mathbf{A}\mathbf{x}$, where the matrix $\mathbf{A} \in \mathbb{R}^{m \times n}$ implements a downsampling operation like bicubic downsampling, or blurring followed by downsampling. As first shown in the pioneering work of Dong et al. (2014) data driven neural networks trained end-to-end outperform classical model-based approaches (Gu et al., 2012; Michaeli & Irani, 2013; Timofte et al., 2013).

Dong et al. (2014) reports that for super-resolution with a simple three-layer CNN, the gains from very large training sets do not seem to be as impressive as in high-level vision problems like image classification. Liang et al. (2021) plot the super-resolution performance of the SOTA SwinIR model as a function of the training set size up to 3600 training examples for a fixed network size. When plotting their results on a logarithmic scale, we observe that the performance improvement follows a power-law. It is unclear, however, whether this power law slows beyond this relatively small number of images.

In this Section, we obtain scaling laws for super-resolution for a U-Net over a wide range of training set and network sizes, similar as for denoising and compressive sensing in the main body.

**Datasets.** We use the same training, validation and test sets as described in Appx. A.1 with training set sizes $N \in [100, 100k]$ images from ImageNet. On top we add two larger training sets of size 300k and 600k. Instead of center cropping the training images to $256 \times 256$ we follow the super-resolution experiments in Liang et al. (2021) and train on images cropped to $128 \times 128$ pixels. We consider super-resolution of factor 2 so the low-resolution images have size $64 \times 64$. The low-resolution images are obtained with the bicubic downsampling function of Python's `PIL.Image` package.

**Model variants and training.** We train the same U-Net model used in Section 3 and described in Appx. A.1 but with only one block per encoder/decoder as this resulted in slightly better results than with two blocks. The network is trained end-to-end to map a coarse reconstruction, obtained through bicubic upsampling, to the residual between the coarse reconstruction and the high-resolution ground truth image.

We vary the number of the channels in the first layer in $\{8, 16, 32, 64, 128, 192, 256, 320, 448, 564\}$, which corresponds to $\{0.007, 0.025, 0.10, 0.40, 1.6, 3.6, 6.4, 10.0, 20.0, 31.2\}$ million network parameters. We do not use any data augmentation, since it is unclear how to account for it in the number of training examples.

We use the $\ell_1$-loss and Adam optimizer with its default settings. For all experiments we find a good initial learning rate with the same annealing strategy as described in Appx. A.1. However, instead of picking the largest learning rate for which the validation loss does not diverge, we pick the second largest, which leads to slightly more stable results. We start the annealing with learning rate of $10^{-5}$. In the few cases, where our heuristic leads to a degenerated training curve, typically due to picking a significantly too small or too large learning rate, starting the annealing with a smaller learning rate of $10^{-6}$ resolves the problem.

We do not put a limit on the amount of compute invested into training. To this end, we deploy an automated learning rate decay that reduces the learning rate by $0.5$ if the validation PSNR has not improved by at least 0.001 for 8 epochs. We stop the training once the validation loss did not improve for two consecutive learning rates. For training sets up to 10000 images we found that batch size of 1 works best. For larger training sets we use a batch size of 10.

For training set sizes up to 10000 we train 3 random seeds and pick the best run. As the variance between runs compared to the gain in performance between different training set sizes decreases with increasing training set size we only run one seed for larger training set sizes.

The experiments were conducted on four NVIDIA A40, four NVIDIA RTX A6000 and four NVIDIA Quadro RTX 6000 GPUs. We measure the time in GPU hours until the best epoch according to the validation loss resulting in about 1500 GPU hours for the experiments in Fig. 6.

**Results and discussion.** For each training set size, Fig. 6(a) shows the reconstruction performance in PSNR of the best model over all simulated network sizes in Fig. 6(b). Since the curves per training set size in Fig. 6(b) are relatively flat, further scaling up the network size is not expected to significantly improve the performance on the studied training sets. Here are the two main findings:

A linear power law with a scaling coefficient $\alpha = 0.0075$ holds roughly up to training set sizes of about 30k images, and for training set sizes starting from 60k this slows to a linear power law with significantly smaller scaling coefficient $\alpha = 0.0029$. With this slowed scaling law a training set size of 1.6B images would be required to increase performance by another 1dB (assuming the relation $32.05N^{0.0029}$ persists, which is likely to slow down even further).

While slowing already at a few tens of thousands of training images, the scaling laws for super-resolution do not slow as early as those for denoising and compressive sensing (see Fig.1). This could be partially due training on image patches of size $128 \times 128$ as opposed to size $256 \times 256$ used for denoising.

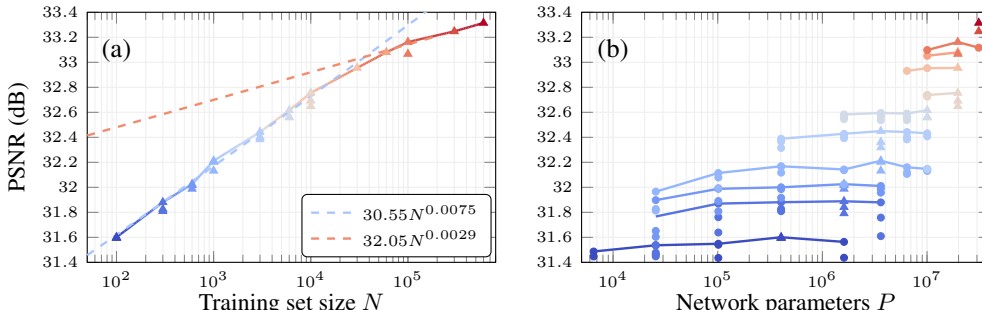

Figure 6: **Empirical scaling laws for super-resolution with a U-Net.** The curve in (a) contains the best reconstruction performances per training set size over the different network sizes depicted in (b). Colors in the plot on the left and right correspond to the same training set size. While the scaling laws for super-resolution do not slow as early as those for Denoising and Compressive sensing (see Fig.1), they are likely to slow further at a larger number of training examples

# D  ADDITIONAL EMPIRICAL SCALING LAWS FOR GAUSSIAN DENOISING

In this Section, we extend our results for Gaussian denoising with a U-Net from Section 3 with two additional setups. Section D.1 considers fixing the noise sampled in each training epoch and Section D.2 investigates reducing the noise level from $\sigma_z = 25$ to $\sigma_z = 15$.

## D.1  EMPIRICAL SCALING LAWS FOR DENOISING WITH FIXED NOISE

Our main results for denoising with a U-Net trained end-to-end discussed in Section 3 follow a setup in which the noise per training example is re-sampled in every training epoch. We choose this setup since it makes the best use of the available clean images. However, fixing the noise is also interesting since it is closer to a denoising setup in which the noise statistics are unknown, which is the case in some real-world noise removal problems. In such a problem, we would be given pairs of noisy and clean image, and could not synthesis new noisy images from the clean images.

In this section, we follow the same experimental setup from Appx. A.1 to simulate the performance of a U-Net for denoising with fixed noise for up to 100k training images (see Fig. 7). Compared to re-sampling the noise, we observe a drop in performance of about 0.3dB for small training set sizes. The performance difference at 10k images is reduced to 0.2dB resulting in a slightly steeper scaling law for moderate training set sizes. However, at around 10k training images the scaling of the performance of training with fixed noise also starts to flatten as it approaches the performance of re-sampling the noise during training. This indicates that if the noise statistics are unknown, more data is required to achieve the same performance as when they are known. However, in both cases the scaling with training set size slows already down at moderate training set sizes.

## D.2  EMPIRICAL SCALING LAWS FOR DENOISING WITH A SMALLER NOISE LEVEL

Our results for Gaussian denoising in Section 3 are for a fixed noise level of $\sigma_z = 25$. In this Section, we repeat the experiments for Gaussian denoising with a U-Net described in Appx. A.1 with smaller noise level of $\sigma_z = 15$, in order to see how the scaling laws change. The results for both noise levels are depicted in Fig. 8.

We observe an improvement of about 2.3dB in PSNR, which is expected since the irreducible error decreases for smaller noise levels. We also observe that the scaling coefficient for the smaller noise level $\sigma_z = 15$ (i.e., $\alpha = 0.0026$) is slightly steeper than that for the larger noise level $\sigma_z = 25$ (i.e., $\alpha = 0.0019$). This coincides with the qualitative behavior of the curves for subspace denoising in Figure 3. Apart from that the curves are qualitatively similar, in that a initially steep power law is replaced by a slower one at around 6000 training images.

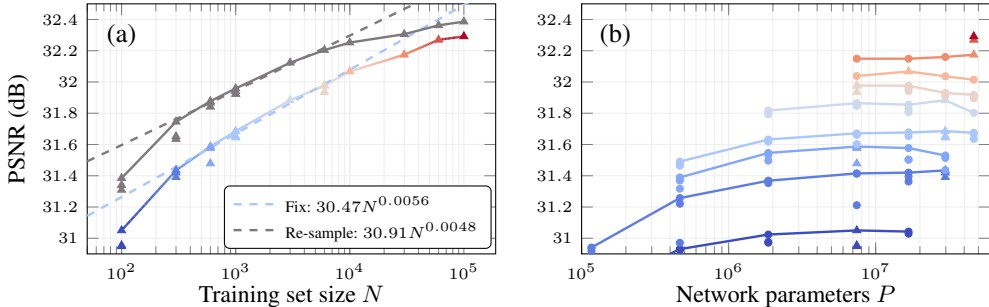

Figure 7: **Empirical scaling laws for denoising with fixed noise.** The colored curve in (a) contains the best reconstruction performances per training set size over the different network sizes depicted in (b) for fixed noise realizations during training. Colors in the plot on the left and right correspond to the same training set size. The gray curve is taken from Fig. 1(a) and shows the performance, when the noise is re-sampled during training. The initial drop in performance due to fixing the noise during training reduces as the training set size increase resulting in a slightly steeper scaling compared to re-sampling the noise. Yet, we expect also the scaling of the performance of training with fixed noise to flatten as it approaches the performance of re-sampling the noise.

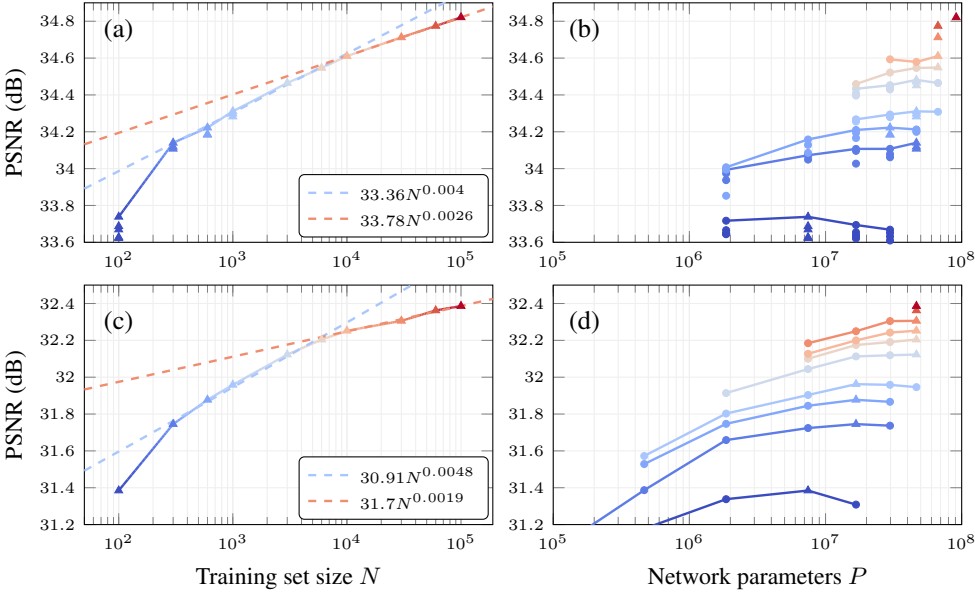

Figure 8: **Comparison of empirical scaling laws for denoising with noise level 15 and 25.** The colored curves in (a) and (c) contain the best reconstruction performances per training set size over the different network sizes depicted in (b) and (d) for noise level 15 and 25 respectively. Colors in the plot on the left and right correspond to the same training set size. The curves in (c),(d) are taken from Fig. 1(a),(b).

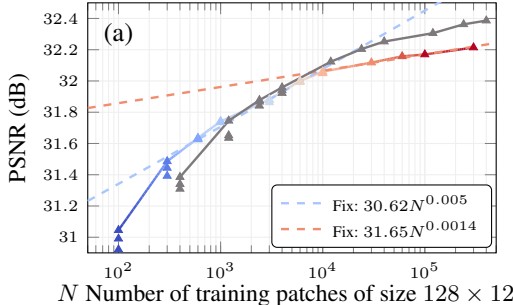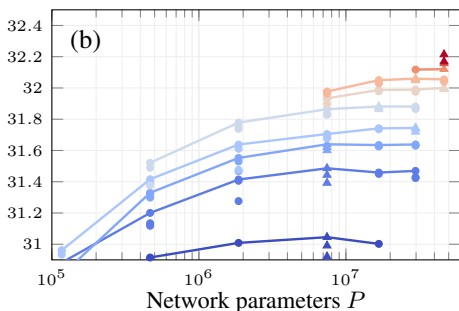

Figure 9: **Empirical scaling laws for denoising with a smaller patch size.** The colored curve in (a) contains the best reconstruction performances per training set size over the different network sizes depicted in (b) for training patches of size $128 \times 128$. Colors in the plot on the left and right correspond to the same training set size. The gray curve is taken from Fig. 1(a) and shows the performance for training patches of size $256 \times 256$. Note that the x-axis shows the number of training patches of size $128 \times 128$. Hence, one patch of size $256 \times 256$ is worth 4 patches of size $128 \times 128$. We see that in the regime of large training set sizes larger patches performs better than more but smaller patches.

### D.3 EMPIRICAL SCALING LAWS FOR DENOISING WITH A SMALLER PATCH SIZE

Our results for Gaussian denoising in Section 3 with a U-Net were obtained for a constant training patch size of $256 \times 256$ pixels across all network and training set sizes. In this Section, we repeat the experiments for Gaussian denoising with a U-Net described in Appx. A.1 with a smaller patch size of $128 \times 128$ pixels.

The results for both patch sizes are depicted in Fig. 9. We observe that in the regime of large training set sizes, that we are primarily interested in, training on $N$ patches of size $256 \times 256$ is more beneficial than training on $4N$ patches of size $128 \times 128$. We therefore focus on patch size $256 \times 256$ in the main body of this work.

### E    UNDERSTANDING SCALING LAWS FOR DENOISING THEORETICALLY - SUPPLEMENTARY RESULTS

In this section, we provide additional details on the statements in Section 5 on understanding scaling laws for denoising theoretically by studying a linear subspace denoising problem theoretically, and provide additional numerical results.

Recall that we consider a linear estimator of the form $f_{\mathbf{W}}(\mathbf{y}) = \mathbf{W}\mathbf{y}$, and measure performance in terms of the expected mean-squared reconstruction error (normalized by the latent signal dimension $d$):

$$R(\mathbf{W}) = \frac{1}{d}\mathbb{E}\left[\|\mathbf{W}\mathbf{y} - \mathbf{x}\|_2^2\right]$$
$$= \frac{1}{d}\|(\mathbf{W} - \mathbf{I})\mathbf{U}\|_F^2 + \frac{\sigma_z^2}{d}\|\mathbf{W}\|_F^2. \tag{1}$$

Above, expectation is over the joint distribution of $(\mathbf{x}, \mathbf{y})$, and the second equality follows from using that $\mathbf{x} = \mathbf{U}\mathbf{c}$, where $\mathbf{c} \sim \mathcal{N}(0, \mathbf{I})$ is Gaussian, and $\mathbf{y} = \mathbf{x} + \mathbf{z}$, where the noise $\mathbf{z} \sim \mathcal{N}(0, \sigma_z^2\mathbf{I})$ is Gaussian.

**The optimal linear estimator.**    The optimal linear estimator (i.e., the estimator that minimizes the risk defined in equation (1)) is given by $\mathbf{W}^* = \frac{1}{1+\sigma_z^2}\mathbf{U}\mathbf{U}^T$. This follows from taking the gradient of the risk (1), setting it to zero, and solving for $\mathbf{W}$. The estimator projects the data onto the subspace and shrinks towards zero, depending on the noise variance. The associated risk is $R(\mathbf{W}^*) = \sigma_z^2/(1 + \sigma_z^2)$.

**Early-stopped empirical risk minimization.** We consider the estimator that applies gradient descent to the empirical risk

$$\mathcal{L}(\mathbf{W}) = \sum_{i=1}^{N} \|\mathbf{W}\mathbf{y}_i - \mathbf{x}_i\|_2^2 = \|\mathbf{W}\mathbf{Y} - \mathbf{X}\|_F^2, \tag{2}$$

where $\mathbf{X}, \mathbf{Y} \in \mathbb{R}^{n \times N}$ contain the training examples as columns, and early-stops after $k$ iterations for regularization.

We next discuss the early-stopped estimator $\mathbf{W}^k$ in more detail. Fig. 10 numerically demonstrates the regularizing effect of early stopping gradient descent, where $\mathbf{W}^\infty = \mathbf{X}\mathbf{Y}^\dagger$ is the converged learned estimator (see Appx. G, Eq. (5)). We see that regularization is necessary for this estimator to perform well.

We next discuss Theorem 2 and the associated assumptions in more detail. Theorem considers the following regime: (i) the number of training examples obeys $(d + n\sigma_z^2)\log(n) \leq N$ and (ii) $N \leq \xi d/\sigma_z^2$, for an arbitrary $\xi$, and (iii) $N \log(N) \leq n$. For this regime, the theorem guarantees that the risk of the optimally early-stopped estimator obeys, with high probability,

$$R(\mathbf{W}^{k_{opt}}) \leq (8 + 2\xi)R(\mathbf{W}^*) + c\left(1 + \frac{n\sigma_z^2}{d}\right)\log n \frac{(d + n\sigma_z^2)\log(n)}{N} + c\xi\sqrt{\frac{(d + \sigma_z^2 n)\log n}{N}}. \tag{3}$$

The theorem looks similar to that for the PCA estimate (Theorem 1), in that the risk is a constant away from the optimal risk, with an error term that becomes small as $(d + n\sigma_z^2)/N$ becomes small. However, the error bound does not converge to $R(\mathbf{W}^*)$ as the number of training examples, $N$, converges to infinity. This is probably an artifact of our analysis, but it is unclear, at least to us, how to derive a substantially tighter bound. In our analysis (see appendix G), we balance two errors: One error decreases in $k$ and is associated with the part of the signal projected into the subspace, and the second error increases with $k$ and is associated with the orthogonal complement of the subspace. We choose the early-stopping time to optimally balance those two terms, which yields the stated bound (3).

Now with regards to the assumption: Assumption (iii) $N \log(N) \leq n$ means we are in the high-dimensional regime; we think this is somewhat closer to reality (for example for denoising a $512 \times 512$ image, this would require the number of training examples to be smaller than 250k), but we can derive an analogous bound for the regime $N \log(N) \geq n$, where the number of training examples is larger than the ambient dimension.

Assumption (i) $(d + n\sigma_z^2)\log(n) \leq N$ is relatively mild, as it is necessary to being able to somewhat accurately estimate the subspace; this assumption is also required for the PCA estimate.

Assumption (ii) $N \leq \xi d/\sigma_z^2$, for an arbitrary $\xi$, is not restrictive in that $\xi$ can be arbitrarily large, we make this assumption only so that the theorem can be stated in a convenient way. However, assumption (ii) reveals a shortcoming of Theorem 2 which is that we cannot make the bound go to zero as $N \to \infty$, since increasing $\xi$ increases one term in the bound, and decreases another one.

### E.1 ADDITIONAL NUMERICAL SIMULATIONS

In this Section we provide further numerical simulations for the PCA subspace estimator and the estimator learned with early stopped gradient descent discussed in Section 5, Theorem 1 and Theorem2. Similar to Fig. 3, Fig. 11 shows the risks $R(\mathbf{W}^{k_{opt}})$ and $R(\mathbf{W}_{\text{PCA}})$ as a function of the number of training examples $N$ for varying values of the signal and ambient dimension $d$ and $n$, while fixing all other model parameters. In the power law region we fit linear power laws with negative scaling coefficients $\alpha$. We observe steeper power laws (larger $|\alpha|$) for smaller ambient dimensions $n$ and larger signal dimensions $d$. Also the scaling coefficients of the learned estimator consistently excel the coefficients from the PCA estimator.

## F PROOF FOR THEOREM 1: RISK BOUND FOR PCA SUBSPACE ESTIMATION

We provide a bound on the risk of the estimator $f(\mathbf{y}) = \mathbf{W}_{\text{PCA}}\mathbf{y}$ with $\mathbf{W}_{\text{PCA}} = \tau\hat{\mathbf{U}}\hat{\mathbf{U}}^T$ and $\tau = \frac{1}{1+\sigma_z^2}$. Recall that $\hat{\mathbf{U}} \in \mathbb{R}^{n \times d}$ contains the singular vectors corresponding to the $d$-leading

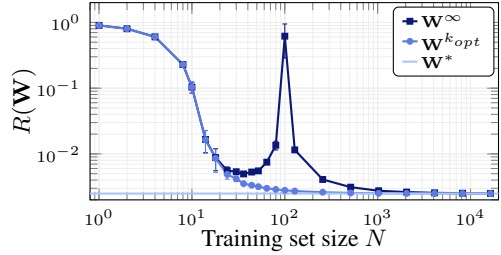

Figure 10: **Effect of early stopping the learned estimator.** The risk of the early stopped learned estimator $\mathbf{W}^{k_{opt}}$ and converged estimator $\mathbf{W}^{\infty}$ as a function of the training set size $N$ measured in simulations. While both estimators approach the optimal performance $R(\mathbf{W}^*)$ for large $N$, early stopping is critical for performance in the regime $N \approx n$. We consider the setup $d = 10, n = 100, \sigma_z = 0.05$. Error bars are over 5 independent runs.

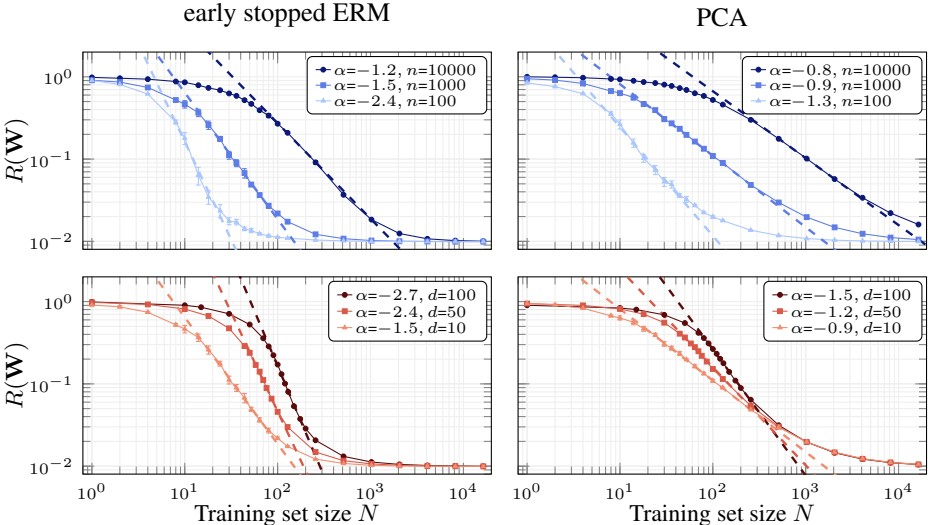

Figure 11: **Additional numerical results for subspace denoising.** Left and right show the simulated risk of the early stopped empirical risk minimizer $\mathbf{W}^{k_{opt}}$ and the PCA estimator $\mathbf{W}_{\mathrm{PCA}}$ as a function of the training set size $N$. In the upper part we fix the signal dimension and noise level $d = 10, \sigma_z = 0.1$ and vary the ambient dimension $n$. In the lower part we fix the ambient dimension and noise level $n = 1000, \sigma_z = 0.1$ and vary the signal dimension $d$. We fit linear scaling laws $\sim N^\alpha$ to the power law regions. Similar to Fig. 3 we observe that with varying model parameters the scaling coefficients $\alpha$ change. Further, the learned estimator exhibits steeper scaling than the PCA estimator over all settings. Error bars are over 5 independent runs.

singular values of $\mathbf{Y}\mathbf{Y}^T$. We define $\hat{\mathbf{U}}_\perp \in \mathbb{R}^{n \times n-d}$ as the orthogonal complement of $\hat{\mathbf{U}}$. Starting from the risk expression given in equation (1) we obtain

$$
\begin{aligned}
R(\mathbf{W}_{\text{PCA}}) &= \frac{1}{d}\left\|(\tau\hat{\mathbf{U}}\hat{\mathbf{U}}^T - \mathbf{I})\mathbf{U}\right\|_F^2 + \frac{1}{d}\tau^2\sigma_z^2\left\|\hat{\mathbf{U}}\hat{\mathbf{U}}^T\right\|_F^2 \\
&= \frac{1}{d}\left\|((\tau-1)\hat{\mathbf{U}}\hat{\mathbf{U}}^T + \hat{\mathbf{U}}_\perp\hat{\mathbf{U}}_\perp^T)\mathbf{U}\right\|_F^2 + \tau^2\sigma_z^2 \\
&= \frac{1}{d}(\tau-1)^2\left\|\hat{\mathbf{U}}^T\mathbf{U}\right\|_F^2 + \frac{1}{d}\left\|\hat{\mathbf{U}}_\perp^T\mathbf{U}\right\|_F^2 + \tau^2\sigma_z^2 \\
&= \frac{1}{d}(\tau-1)^2\left(d - \left\|\hat{\mathbf{U}}_\perp^T\mathbf{U}\right\|_F^2\right) + \frac{1}{d}\left\|\hat{\mathbf{U}}_\perp^T\mathbf{U}\right\|_F^2 + \tau^2\sigma_z^2 \\
&= (1 - (\tau-1)^2)\frac{1}{d}\left\|\hat{\mathbf{U}}_\perp^T\mathbf{U}\right\|_F^2 + (\tau-1)^2 + \sigma_z^2\tau^2 \\
&= \frac{1 + 2\sigma_z^2}{(1+\sigma_z^2)^2}\frac{1}{d}\left\|\hat{\mathbf{U}}_\perp^T\mathbf{U}\right\|_F^2 + \frac{\sigma_z^2}{1+\sigma_z^2} \\
&\overset{(i)}{\leq} \frac{1 + 2\sigma_z^2}{(1+\sigma_z^2)^2}\left\|\hat{\mathbf{U}}_\perp^T\mathbf{U}\right\|^2 + \frac{\sigma_z^2}{1+\sigma_z^2} \\
&\overset{(ii)}{\leq} c\frac{1 + 2\sigma_z^2}{(1+\sigma_z^2)^2}\frac{(d+n\sigma_z^2)\log(2n)}{N} + \frac{\sigma_z^2}{1+\sigma_z^2} \\
&\leq c\frac{(d+n\sigma_z^2)\log(n)}{N} + \frac{\sigma_z^2}{1+\sigma_z^2}.
\end{aligned}
\tag{4}
$$

Here, inequality (ii) follows from Section H.1 equation (34) and holds in the regime $(d + n\sigma_z^2)\log(n) \leq N$, for some constant $c$ and with probability at least $1 - n^{-10} - 3e^{-d} + e^{-n}$. This concludes the proof.

## G PROOF OF THEOREM 2: RISK BOUND FOR EARLY STOPPED EMPIRICAL RISK MINIMIZATION

This section contains the proof of Theorem 2. The theorem characterizes the performance of the learned, linear estimator $\mathbf{W}^k$ that is obtained by applying $k$ iterations of gradient descent with stepsize $\eta$ starting at $\mathbf{W}^0 = 0$ to the loss $\mathcal{L}(\mathbf{W})$ in equation (2).

We start by deriving a closed form expression for the estimator $\mathbf{W}^k$. The gradient of the loss is

$$\nabla_{\mathbf{W}}\mathcal{L}(\mathbf{W}) = (\mathbf{W}\mathbf{Y} - \mathbf{X})\mathbf{Y}^T,$$

and thus the iterations of gradient descent are

$$
\begin{aligned}
\mathbf{W}^{k+1} &= \mathbf{W}^k - \eta(\mathbf{W}^k\mathbf{Y} - \mathbf{X})\mathbf{Y}^T \\
&= \mathbf{W}^k(\mathbf{I} - \eta\mathbf{Y}\mathbf{Y}^T) + \eta\mathbf{X}\mathbf{Y}^T.
\end{aligned}
$$

Let $\mathbf{Y} = \mathbf{U}_y\boldsymbol{\Sigma}_y\mathbf{V}_y^T \in \mathbb{R}^{n \times N}$ and $\mathbf{X} = \mathbf{U}_x\boldsymbol{\Sigma}_x\mathbf{V}_x^T \in \mathbb{R}^{n \times N}$ be the singular value decompositions of $\mathbf{Y}$ and $\mathbf{X}$ respectively and assume that the singular values are non-zero and descending, i.e., $\sigma_{y,1} \geq \sigma_{y,2} \geq \dots$. We have, with $\mathbf{W}_0 = 0$ and $k \geq 1$ that

$$
\begin{aligned}
\mathbf{W}^k &= \eta\mathbf{X}\mathbf{Y}^T\sum_{\ell=0}^{k-1}(\mathbf{I} - \eta\mathbf{Y}\mathbf{Y}^T)^\ell \\
&= \eta\mathbf{X}\mathbf{V}_y\boldsymbol{\Sigma}_y\mathbf{U}_y^T\left(\sum_{\ell=0}^{k-1}(\mathbf{I} - \eta\mathbf{U}_y\boldsymbol{\Sigma}_y^2\mathbf{U}_y^T)^\ell\right) \\
&= \eta\mathbf{X}\mathbf{V}_y\boldsymbol{\Sigma}_y\text{diag}\left(\sum_{\ell=0}^{k-1}(1 - \eta\sigma_{y,i}^2)^\ell\right)\mathbf{U}_y^T \\
&= \mathbf{X}\mathbf{V}_y\mathbf{D}_k\mathbf{U}_y^T,
\end{aligned}
$$

where we defined $\mathbf{D}_k \in \mathbb{R}^{N \times N}$ as a diagonal matrix with $i$-th diagonal entry given by $(1 - (1 - \eta\sigma_{y,i}^2)^k)/\sigma_{y,i}$ and where we used the geometric series to obtain

$$\sum_{\ell=0}^{k-1}(1 - \eta\sigma_{y,i}^2)^\ell = \frac{1 - (1 - \eta\sigma_{y,i}^2)^k}{\eta\sigma_{y,i}^2}.$$

Note that for $k \to \infty$ and choosing $\eta$ such that $1 - \eta\sigma_{y,i}^2 < 1$ for all $i$ we get

$$\mathbf{W}^\infty = \mathbf{X}\mathbf{V}_y\mathbf{\Sigma}_y^{-1}\mathbf{U}_y^T = \mathbf{X}\mathbf{Y}^\dagger. \tag{5}$$

Evaluating the risk from equation (1) at the estimator $\mathbf{W}^k$ gives

$$R(\mathbf{W}^k) = \frac{1}{d}\left\|(\mathbf{X}\mathbf{V}_y\mathbf{D}_k\mathbf{U}_y^T - \mathbf{I})\mathbf{U}\right\|_F^2 + \frac{\sigma_z^2}{d}\left\|\mathbf{X}\mathbf{V}_y\mathbf{D}_k\mathbf{U}_y^T\right\|_F^2. \tag{6}$$

To shorten notation we define

$$\gamma := \frac{(d + n\sigma_z^2)\log(n)}{N} \tag{7}$$

$$\psi := \frac{n\sigma_z^2\log(n)}{N}. \tag{8}$$

We next provide bounds for the two terms on the right-hand-side of equation (6), proven later in this section.

**Bound on the first term in equation** (6): In Section G.1 we show that provided $N\log(N) \leq n$, $9d \leq N$ and $(d + n\sigma_z^2)\log(n) \leq N$, for some constant $c$, with probability at least $1 - 2e^{-N/18} - 3n^{-10} - 3e^{-d} - e^{-n} - e^{-N} - 2e^{-n/2}$, the following bound holds:

$$\frac{1}{d}\left\|(\mathbf{X}\mathbf{V}_y\mathbf{D}_k\mathbf{U}_y^T - \mathbf{I})\mathbf{U}\right\|_F^2 \leq \frac{c}{d}\left(\sigma_z^2 n + \gamma\sigma_{x,max}^2\right)\sum_{i=1}^{d}\frac{1}{\sigma_{y,i}^2}(1 - (1 - \eta\sigma_{y,i}^2)^k)^2 + \frac{c}{d}\sum_{i=1}^{d}(1 - \eta\sigma_{y,i}^2)^{2k}$$

$$+ \frac{c}{d}\psi\gamma\sum_{i=d+1}^{N}\frac{\sigma_{x,max}^2}{\sigma_{y,i}^2}(1 - (1 - \eta\sigma_{y,i}^2)^k)^2 + c\gamma. \tag{9}$$

**Bound on the second term in equation** (6): In Section G.2 we show

$$\frac{\sigma_z^2}{d}\left\|\mathbf{X}\mathbf{V}_y\mathbf{D}_k\mathbf{U}_y^T\right\|_F^2 \leq \frac{\sigma_z^2}{d}\sum_{i=1}^{N}\frac{\sigma_{x,max}^2}{\sigma_{y,i}^2}(1 - (1 - \eta\sigma_{y,i}^2)^k)^2. \tag{10}$$

With equations (9) and (10) in place and by splitting up the sum in (10) we can bound the right hand side of equation (6) as

$$R(\mathbf{W}^k) \leq \frac{1}{d}\left(c\sigma_z^2 n + c\gamma\sigma_{x,max}^2 + \sigma_z^2\sigma_{x,max}^2\right)\sum_{i=1}^{d}\frac{1}{\sigma_{y,i}^2}(1 - (1 - \eta\sigma_{y,i}^2)^k)^2 + \frac{c}{d}\sum_{i=1}^{d}(1 - \eta\sigma_{y,i}^2)^{2k}$$

$$+ c\gamma + \frac{c}{d}\left(\psi\gamma + \sigma_z^2\right)\sum_{i=d+1}^{N}\frac{\sigma_{x,max}^2}{\sigma_{y,i}^2}(1 - (1 - \eta\sigma_{y,i}^2)^k)^2. \tag{11}$$

Since the singular values of $\mathbf{Y}$ are in descending order $\sigma_{y,1} \geq \sigma_{y,2} \geq \ldots$, this is, for any iteration $k$, bounded as

$$R(\mathbf{W}^k) \leq \left(c\sigma_z^2 n + c\gamma\sigma_{x,max}^2 + \sigma_z^2\sigma_{x,max}^2\right)\frac{1}{\sigma_{y,d}^2} + c(1 - \eta\sigma_{y,d}^2)^{2k}$$

$$+ c\gamma + \frac{c}{d}\left(\psi\gamma + \sigma_z^2\right)\sigma_{x,max}^2\sum_{i=d+1}^{N}\frac{1}{\sigma_{y,i}^2}(1 - (1 - \eta\sigma_{y,i}^2)^k)^2. \tag{12}$$

This is a good bound if we're at an iteration sufficiently large so that $(1 - \eta\sigma_{y,1}^2)^k$ is small.

In (12) we have $(1 - \eta\sigma_{y,d}^2)^{2k}$ that is decreasing in the number of gradient descent steps $k$ and also decreasing in the stepsize $\eta$ as long as $\eta\sigma_{y,i}^2 \le 1$ for $i = 1, \ldots, d$. Further, we have $(1 - (1 - \eta\sigma_{y,i}^2)^k)^2$ that is increasing in $k$ and also increasing in $\eta$. The first term corresponds to the signal that we want to fit sufficiently well, whereas the second term corresponds to the noise from which we want to fit as little as possible. Hence, there exist optimal choices for $k, \eta$ that trade-off the sum of the two terms.

In our setup the $d$ leading singular values of $\mathbf{Y}$ corresponding to the signal are large and concentrate around $N(1 + \sigma_z^2)$, while the remaining singular values are small and concentrate around $N\sigma_z^2$. Hence, we can apply a single step of gradient descent $k = 1$ to already fit a large portion of the signal, while minimizing the portion of the noise that is fitted. For that we choose the stepsize $\eta$ as large as possible such that $\eta\sigma_{y,i}^2 \le 1$ for $i = 1, \ldots, d$ still holds.

Next, suppose the following events hold

$$\mathcal{E}_1 = \{\sigma_{y,d+1}^2 \le N\left(\sigma_z^2 + \epsilon(1 + \sigma_z^2)\right)\} \tag{13}$$

$$\mathcal{E}_2 = \{\sigma_{y,d}^2 \ge N(1 + \sigma_z^2)(1 - \epsilon)\} \tag{14}$$

$$\mathcal{E}_3 = \{\sigma_{x,\max}^2 \le 4N\} \tag{15}$$

$$\mathcal{E}_4 = \{\sigma_{y,1}^2 \le N(1 + \epsilon)(1 + \sigma_z^2)\}. \tag{16}$$

In Section H.4, we show that

$$\mathrm{P}\left[\mathcal{E}_3\right] \ge 1 - 2e^{-N/8}. \tag{17}$$

In Section H.4, we also show that, provided that $N \ge 3C\epsilon^{-2}(d + \sigma_z^2 n)\log n$, for some constant $C$ and $\epsilon \in (0, 1)$,

$$\mathrm{P}\left[\mathcal{E}_1\right], \mathrm{P}\left[\mathcal{E}_2\right], \mathrm{P}\left[\mathcal{E}_4\right] \ge 1 - e^{-d} - e^{-n} - n^{-9}. \tag{18}$$

We next bound the terms in equation (12). As discussed, we set $k = 1$ and the stepsize as large as possible, i.e. $\eta = 1/\left(N(1 + \epsilon)(1 + \sigma_z^2)\right) \le 1/\sigma_{y,1}^2$, which holds on event $\mathcal{E}_4$. Finally, on event $\mathcal{E}_2$ we obtain

$$c(1 - \eta\sigma_{y,d}^2)^{2k} \le c(1 - \eta N(1 + \sigma_z^2)(1 - \epsilon))^2 = c\left(1 - \frac{1 - \epsilon}{1 + \epsilon}\right)^2 \le c\epsilon^2.$$

Next we bound the sum in equation (12). Towards this goal, we upper bound each term in the sum with its linear approximation at the origin. We compute the derivative at the origin as

$$\lim_{q \to 0} \frac{\partial}{\partial q} \frac{1}{q} (1 - (1 - \eta q)^k)^2 = (\eta k)^2.$$

Thus, we have

$$\sum_{i=d+1}^{N} \frac{1}{\sigma_{y,i}^2}(1 - (1 - \eta\sigma_{y,i}^2)^k)^2 \le \sum_{i=d+1}^{N} k^2\eta^2\sigma_{y,i}^2 \le Nk^2\eta^2\sigma_{y,d+1}^2 \le \frac{\sigma_z^2 + \epsilon + \epsilon\sigma_z^2}{(1 + \epsilon)^2(1 + \sigma_z^2)^2} \le c(\sigma_z^2 + 2\epsilon),$$

on the event $\mathcal{E}_1$ and for $\sigma_z^2 \le 1$, $k = 1$ and $\eta = 1/\left(N(1 + \epsilon)(1 + \sigma_z^2)\right)$. Putting this together and on the events $\mathcal{E}_2, \mathcal{E}_3$ we get the bound

$$R(\mathbf{W}^k) \le 8\frac{\sigma_z^2}{1 + \sigma_z^2} + c\gamma + c(1 - \eta\sigma_{y,d}^2)^{2k} + \frac{c}{d}\left(\psi\gamma + \sigma_z^2\right)N\sum_{i=d+1}^{N}\frac{1}{\sigma_{y,i}^2}(1 - (1 - \eta\sigma_{y,i}^2)^k)^2$$

$$\le 8\frac{\sigma_z^2}{1 + \sigma_z^2} + c\gamma + c\epsilon^2 + \frac{cN}{d}\left(\frac{\sigma_z^2 n \log n}{N}\frac{(d + n\sigma_z^2)\log n}{N} + \sigma_z^2\right)(\sigma_z^2 + 2\epsilon)$$

$$\le 8R(\mathbf{W}^*) + c\frac{(d + n\sigma_z^2)\log(n)}{N} + c\epsilon^2 + c\left(\frac{((d + n\sigma_z^2)\log n)^2}{dN} + \frac{\sigma_z^2 N}{d}\right)(\sigma_z^2 + 2\epsilon). \tag{19}$$

The bound holds provided $3C\epsilon^{-2}(d + \sigma_z^2 n)\log n \le N$ and $N\log(N) \le n$, for some constants $c, C$ and with probability at least $1 - 2e^{-N/8} - 2e^{-N/18} - 5n^{-9} - 5e^{-d} - 2e^{-n} - e^{-N} - 2e^{-n/2}$.

To maximize the benefit from early stopping we set $\epsilon$ as small as possible with respect to the condition $3C\epsilon^{-2}(d + \sigma_z^2 n)\log n \le N$

$$\epsilon = \sqrt{\frac{3C(d + \sigma_z^2 n)\log n}{N}}. \tag{20}$$

With that equation (19) becomes

$$R(\mathbf{W}^k) \le 8R(\mathbf{W}^*) + c\frac{(d + n\sigma_z^2)\log(n)}{N} + c\left(\frac{((d + n\sigma_z^2)\log n)^2}{dN} + \frac{\sigma_z^2 N}{d}\right)\left(\sigma_z^2 + \sqrt{\frac{(d + \sigma_z^2 n)\log n}{N}}\right)$$

$$= 8R(\mathbf{W}^*) + c\gamma + c\left(\gamma\left(1 + \frac{n\sigma_z^2}{d}\right)\log n + \frac{\sigma_z^2 N}{d}\right)\left(\sigma_z^2 + \sqrt{\gamma}\right)$$

$$\le 8R(\mathbf{W}^*) + c\gamma\left(1 + \frac{n\sigma_z^2}{d}\right)\log n + c\frac{\sigma_z^2 N}{d}\left(\sigma_z^2 + \sqrt{\gamma}\right). \tag{21}$$

We now consider the regime where $N \le \xi\frac{d}{\sigma_z^2}$, for a numerical constant $\xi$, to simplify the statement further. For this regime, we have

$$R(\mathbf{W}^k) \le R(\mathbf{W}^*)(8 + 2\xi) + c\gamma\left(1 + \frac{n\sigma_z^2}{d}\right)\log n + c\xi\sqrt{\gamma}.$$

This concludes the proof of Theorem 2.

### G.1 PROOF OF EQUATION (9)

In this Section, we derive a bound for $\frac{1}{d}\left\|(\mathbf{X}\mathbf{V}_y\mathbf{D}_k\mathbf{U}_y^T - \mathbf{I})\mathbf{U}\right\|_F^2$ the first term in (6). To this end, we introduce further notation. Recall that $\mathbf{Y} = \mathbf{U}_y\mathbf{\Sigma}_y\mathbf{V}_y^T \in \mathbb{R}^{n \times N}$ and $\mathbf{X} = \mathbf{U}_x\mathbf{\Sigma}_x\mathbf{V}_x^T \in \mathbb{R}^{n \times N}$ are the SVDs of $\mathbf{Y}$ and $\mathbf{X}$ respectively. All derivations below hold regardless of whether $N \le n$ or $N > n$. Exemplarily, we will show them for $N \le n$. Thus, $\mathbf{U}_y \in \mathbb{R}^{n \times N}$, $\mathbf{\Sigma}_y \in \mathbb{R}^{N \times N}$, $\mathbf{V}_y \in \mathbb{R}^{N \times N}$ and $\mathbf{U}_x \in \mathbb{R}^{n \times d}$, $\mathbf{\Sigma}_x \in \mathbb{R}^{d \times d}$, $\mathbf{V}_x \in \mathbb{R}^{N \times d}$.

Let $\mathbf{U}_{y1}, \mathbf{U}_{x1} \in \mathbb{R}^{n \times d}$ be the $d$ leading left singular vectors of $\mathbf{Y}$ and $\mathbf{X}$ (note that $\mathbf{U}_{x1} = \mathbf{U}_x$), let $\mathbf{U}_{y2}, \mathbf{U}_{x2} \in \mathbb{R}^{n \times n-d}$ be their orthonormal complements. Let $\tilde{\mathbf{U}}_y = [\mathbf{U}_{y1} \quad \mathbf{U}_{y2}] \in \mathbb{R}^{n \times n}$ and $\tilde{\mathbf{U}}_x = [\mathbf{U}_{x1} \quad \mathbf{U}_{x2}] \in \mathbb{R}^{n \times n}$. Analogous definitions can be made for the leading right singular vectors of $\mathbf{Y}$ and $\mathbf{X}$.

Recall that $\mathbf{D}_k = \text{diag}\left(\ldots, (1 - (1 - \eta\sigma_{y,i}^2)^k)/\sigma_{y,i}, \ldots\right) \in \mathbb{R}^{N \times N}$. Let $\tilde{\mathbf{D}}_k = [\mathbf{D}_k \quad \mathbf{0}] \in \mathbb{R}^{N \times n}$ and define $\tilde{\mathbf{D}}_{k1} \in \mathbb{R}^{d \times d}$ and $\tilde{\mathbf{D}}_{k2} \in \mathbb{R}^{N-d \times n-d}$ such that

$$\tilde{\mathbf{D}}_k = \begin{bmatrix} \tilde{\mathbf{D}}_{k1} & \mathbf{0} \\ \mathbf{0} & \tilde{\mathbf{D}}_{k2} \end{bmatrix}. \tag{22}$$

With these definitions in place we can write

$$\left\|(\mathbf{X}\mathbf{V}_y\mathbf{D}_k\mathbf{U}_y^T - \mathbf{I})\mathbf{U}\right\|_F = \left\|\tilde{\mathbf{U}}_y^T(\mathbf{X}\mathbf{V}_y\tilde{\mathbf{D}}_k\tilde{\mathbf{U}}_y^T - \mathbf{I})\tilde{\mathbf{U}}_y\tilde{\mathbf{U}}_y^T\mathbf{U}\right\|_F$$

$$= \left\|(\tilde{\mathbf{U}}_y^T\mathbf{X}\mathbf{V}_y\tilde{\mathbf{D}}_k - \mathbf{I})\tilde{\mathbf{U}}_y^T\mathbf{U}\right\|_F$$

$$= \left\|\left(\begin{bmatrix} \mathbf{U}_{y1}^T\mathbf{X}\mathbf{V}_{y1}\tilde{\mathbf{D}}_{k1} & \mathbf{U}_{y1}^T\mathbf{X}\mathbf{V}_{y2}\tilde{\mathbf{D}}_{k2} \\ \mathbf{U}_{y2}^T\mathbf{X}\mathbf{V}_{y1}\tilde{\mathbf{D}}_{k1} & \mathbf{U}_{y2}^T\mathbf{X}\mathbf{V}_{y2}\tilde{\mathbf{D}}_{k2} \end{bmatrix} - \begin{bmatrix} \mathbf{I} & 0 \\ 0 & \mathbf{I} \end{bmatrix}\right)\begin{bmatrix} \mathbf{U}_{y1}^T\mathbf{U} \\ \mathbf{U}_{y2}^T\mathbf{U} \end{bmatrix}\right\|_F$$

$$\le \left\|(\mathbf{U}_{y1}^T\mathbf{X}\mathbf{V}_{y1}\tilde{\mathbf{D}}_{k1} - \mathbf{I})\mathbf{U}_{y1}^T\mathbf{U}\right\|_F$$

$$+ \left\|\mathbf{U}_{y2}^T\mathbf{X}\mathbf{V}_{y1}\tilde{\mathbf{D}}_{k1}\mathbf{U}_{y1}^T\mathbf{U}\right\|_F$$

$$+ \left\|\left(\mathbf{U}_y^T\mathbf{X}\mathbf{V}_{y2}\tilde{\mathbf{D}}_{k2} - \begin{bmatrix} 0 \\ \mathbf{I} \end{bmatrix}\right)\mathbf{U}_{y2}^T\mathbf{U}\right\|_F. \tag{23}$$

The first term in (23) is bounded in the regime $n \geq N$, for some constant $c$ and with probability at least $1 - 2e^{-n/2}$ by

$$\left\| (\mathbf{U}_{y1}^T \mathbf{X} \mathbf{V}_{y1} \tilde{\mathbf{D}}_{k1} - \mathbf{I}) \mathbf{U}_{y1}^T \mathbf{U} \right\|_F \leq c\sigma_z \sqrt{n} \sqrt{\sum_{i=1}^d \frac{1}{\sigma_{y,i}^2}(1 - (1 - \eta\sigma_{y,i}^2)^k)^2} + \sqrt{\sum_{i=1}^d (1 - \eta\sigma_{y,i}^2)^{2k}}. \tag{24}$$

See Section G.1.1 for a proof. The second term in (23) is bounded for some constant $c$, with probability at least $1 - 2n^{-10} - 2e^{-d} - e^{-n}$ and in regime $(d + n\sigma_z^2)\log(n) \leq N$ by

$$\left\| \mathbf{U}_{y2}^T \mathbf{X} \mathbf{V}_{y1} \tilde{\mathbf{D}}_{k1} \mathbf{U}_{y1}^T \mathbf{U} \right\|_F \leq c\sqrt{\gamma} \sqrt{\sum_{i=1}^d \frac{\sigma_{x,max}^2}{\sigma_{y,i}^2}(1 - (1 - \eta\sigma_{y,i}^2)^k)^2}. \tag{25}$$

See Section G.1.2 for a proof. The third term in (23) is bounded for some constant $c$, with probability at least $1 - 2e^{-N/18} - 3n^{-10} - 3e^{-d} - e^{-n} - e^{-N}$ and in the regime $(d + n\sigma_z^2)\log(n) \leq N$, $N\log(N) \leq n$ and $9d \leq N$ by

$$\left\| \left( \mathbf{U}_y^T \mathbf{X} \mathbf{V}_{y2} \tilde{\mathbf{D}}_{k2} - \begin{bmatrix} 0 \\ \mathbf{I} \end{bmatrix} \right) \mathbf{U}_{y2}^T \mathbf{U} \right\|_F \leq c\sqrt{\psi\gamma \sum_{i=d+1}^N \frac{\sigma_{x,max}^2}{\sigma_{y,i}^2}(1 - (1 - \eta\sigma_{y,i}^2)^k)^2} + c\sqrt{d\gamma}. \tag{26}$$

See Section G.1.3 for a proof.

Combining those results we can bound equation (23) in the regime $N\log(N) \leq n$, $9d \leq N$ and $(d + n\sigma_z^2)\log(n) \leq N$, for some constant $c$ and with probability at least $1 - 2e^{-N/18} - 3n^{-10} - 3e^{-d} - e^{-n} - e^{-N} - 2e^{-n/2}$ as

$$\frac{1}{d} \left\| (\mathbf{X}\mathbf{V}_y \mathbf{D}_k \mathbf{U}_y^T - \mathbf{I})\mathbf{U} \right\|_F^2 \leq \left( \frac{c\sigma_z^2 n}{d} + \frac{c}{d}\sigma_{x,max}^2 \gamma \right) \sum_{i=1}^d \frac{1}{\sigma_{y,i}^2}(1 - (1 - \eta\sigma_{y,i}^2)^k)^2 + \frac{c}{d}\sum_{i=1}^d (1 - \eta\sigma_{y,i}^2)^{2k}$$

$$+ \frac{c}{d}\psi\gamma \sum_{i=d+1}^N \frac{\sigma_{x,max}^2}{\sigma_{y,i}^2}(1 - (1 - \eta\sigma_{y,i}^2)^k)^2 + c\gamma. \tag{27}$$

This concludes the proof of equation (9).

### G.1.1 PROOF OF EQUATION (24)

In the regime $n \geq N$, for some constant $c$ and with probability at least $1 - 2e^{-n/2}$, the first term in (23) is bounded by

$$\left\| (\mathbf{U}_{y1}^T \mathbf{X} \mathbf{V}_{y1} \tilde{\mathbf{D}}_{k1} - \mathbf{I})\mathbf{U}_{y1}^T \mathbf{U} \right\|_F \overset{(i)}{\leq} \left\| \mathbf{U}_{y1}^T \mathbf{U}_{x1} \mathbf{\Sigma}_{x1} \mathbf{V}_{x1}^T \mathbf{V}_{y1} \tilde{\mathbf{D}}_{k1} - \mathbf{I} \right\|_F$$

$$\leq \left\| \mathbf{U}_{y1}^T \mathbf{U}_{x1} \mathbf{\Sigma}_{x1} \mathbf{V}_{x1}^T \mathbf{V}_{y1} \tilde{\mathbf{D}}_{k1} - \mathbf{\Sigma}_{y1} \tilde{\mathbf{D}}_{k1} \right\| + \left\| \mathbf{\Sigma}_{y1} \tilde{\mathbf{D}}_{k1} - \mathbf{I} \right\|_F$$

$$\leq \left\| \mathbf{U}_{y1}^T \mathbf{U}_{x1} \mathbf{\Sigma}_{x1} \mathbf{V}_{x1}^T \mathbf{V}_{y1} - \mathbf{\Sigma}_{y1} \right\| \left\| \tilde{\mathbf{D}}_{k1} \right\|_F + \left\| \mathbf{\Sigma}_{y1} \tilde{\mathbf{D}}_{k1} - \mathbf{I} \right\|_F$$

$$\overset{(ii)}{\leq} c\sigma_z \sqrt{n} \sqrt{\sum_{i=1}^d \frac{1}{\sigma_{y,i}^2}(1 - (1 - \eta\sigma_{y,i}^2)^k)^2} + \sqrt{\sum_{i=1}^d (1 - \eta\sigma_{y,i}^2)^{2k}}. \tag{28}$$

Inequality (i) uses $\left\| \mathbf{U}_{y1}^T \mathbf{U} \right\| \leq 1$. To obtain inequality (ii), we used that $\tilde{\mathbf{D}}_{k1} = \text{diag}(\ldots, (1 - (1 - \eta\sigma_{y,i}^2)^k)/\sigma_{y,i}, \ldots) \in \mathbb{R}^{d \times d}$ and that

$$\left\| \mathbf{U}_{y1}^T \mathbf{U}_{x1} \mathbf{\Sigma}_{x1} \mathbf{V}_{x1}^T \mathbf{V}_{y1} - \mathbf{\Sigma}_{y1} \right\| \leq \left\| \mathbf{U}_{x1} \mathbf{\Sigma}_{x1} \mathbf{V}_{x1}^T - \mathbf{U}_{y1} \mathbf{\Sigma}_{y1} \mathbf{V}_{y1}^T \right\|$$

$$\leq \left\| \mathbf{U}_x \mathbf{\Sigma}_x \mathbf{V}_x^T - \mathbf{U}_y \mathbf{\Sigma}_y \mathbf{V}_y^T \right\|$$

$$= \left\| \mathbf{Z} \right\|$$

$$\leq c\sigma_z \sqrt{n}. \tag{29}$$

Here, the last inequality holds in the regime $n \geq N$, for some constant $c$ and with probability at least $1 - 2e^{-n/2}$ and follows from Section H.4, equation (64).

### G.1.2 PROOF OF EQUATION (25)

For some constant $c$, with probability at least $1 - 2n^{-10} - 4e^{-d} - e^{-n}$ and in regime $(d + n\sigma_z^2)\log(n) \leq N$ the second term in (23) is bounded by

$$\left\|\mathbf{U}_{y2}^T\mathbf{X}\mathbf{V}_{y1}\tilde{\mathbf{D}}_{k1}\mathbf{U}_{y1}^T\mathbf{U}\right\|_F \leq \left\|\mathbf{U}_{y2}^T\mathbf{U}_{x1}\right\|\left\|\mathbf{V}_{x1}^T\mathbf{V}_{y1}\right\|\left\|\mathbf{U}_{y1}^T\mathbf{U}\right\|\left\|\mathbf{\Sigma}_{x1}\right\|\left\|\tilde{\mathbf{D}}_{k1}\right\|_F$$

$$\overset{(i)}{\leq}\left\|\mathbf{U}_{y2}^T\mathbf{U}_{x1}\right\|\left\|\mathbf{\Sigma}_{x1}\right\|\left\|\tilde{\mathbf{D}}_{k1}\right\|_F$$

$$\overset{(ii)}{\leq}c\sqrt{\gamma}\sqrt{\sum_{i=1}^{d}\frac{\sigma_{x,max}^2}{\sigma_{y,i}^2}(1-(1-\eta\sigma_{y,i}^2)^k)^2}. \tag{30}$$

Here, inequality (i) follows from $\left\|\mathbf{V}_{x1}^T\mathbf{V}_{y1}\right\| \leq 1$ and $\left\|\mathbf{U}_{y1}^T\mathbf{U}\right\| \leq 1$. For inequality (ii) we used that $\tilde{\mathbf{D}}_{k1} = \text{diag}\left(\ldots, (1-(1-\eta\sigma_{y,i}^2)^k)/\sigma_{y,i}, \ldots\right) \in \mathbb{R}^{d\times d}$ and the bound on $\left\|\mathbf{U}_{y2}^T\mathbf{U}_{x1}\right\|$ from Section H.1, equation (35) that holds in the regime $(d + n\sigma_z^2)\log(n) \leq N$, for some constant $c$ and with probability at least $1 - 2n^{-10} - 2e^{-d} - e^{-n}$.

### G.1.3 PROOF OF EQUATION (26)

For some constant $c$, with probability at least $1 - 2e^{-N/18} - 3n^{-10} - 3e^{-d} - e^{-n} - e^{-N}$ and in the regime $(d + n\sigma_z^2)\log(n) \leq N$ and $9d \leq N$ we obtain

$$\left\|\left(\mathbf{U}_y^T\mathbf{X}\mathbf{V}_{y2}\tilde{\mathbf{D}}_{k2} - \begin{bmatrix}0\\\mathbf{I}\end{bmatrix}\right)\mathbf{U}_{y2}^T\mathbf{U}\right\|_F = \left\|\begin{bmatrix}\mathbf{U}_{y1}^T\mathbf{X}\mathbf{V}_{y2}\tilde{\mathbf{D}}_{k2}\mathbf{U}_{y2}^T\mathbf{U}\\\mathbf{U}_{y2}^T\mathbf{X}\mathbf{V}_{y2}\tilde{\mathbf{D}}_{k2}\mathbf{U}_{y2}^T\mathbf{U} - \mathbf{U}_{y2}^T\mathbf{U}\end{bmatrix}\right\|_F$$

$$\leq\left\|\mathbf{U}_{y1}^T\mathbf{U}_{x1}\mathbf{\Sigma}_{x1}\mathbf{V}_{x1}^T\mathbf{V}_{y2}\tilde{\mathbf{D}}_{k2}\mathbf{U}_{y2}^T\mathbf{U}\right\|_F$$

$$+\left\|\mathbf{U}_{y2}^T\mathbf{U}_{x1}\mathbf{\Sigma}_{x1}\mathbf{V}_{x1}^T\mathbf{V}_{y2}\tilde{\mathbf{D}}_{k2}\mathbf{U}_{y2}^T\mathbf{U}\right\|_F$$

$$+\sqrt{d}\left\|\mathbf{U}_{y2}^T\mathbf{U}\right\|$$

$$\overset{(i)}{\leq}(\sqrt{\gamma}+\gamma)c\sqrt{\psi}\left\|\mathbf{\Sigma}_{x1}\right\|\left\|\tilde{\mathbf{D}}_{k2}\right\|_F + c\sqrt{d\gamma}$$

$$\overset{(ii)}{\leq}c\sqrt{\psi\gamma\sum_{i=d+1}^{N}\frac{\sigma_{x,max}^2}{\sigma_{y,i}^2}(1-(1-\eta\sigma_{y,i}^2)^k)^2} + c\sqrt{d\gamma}. \tag{31}$$

Here, inequality (i) follows from $\left\|\mathbf{U}_{y1}^T\mathbf{U}_{x1}\right\| \leq 1$ and the bound in Section H.1, equations (34), (35) and (36) and holds in the regime $(d + n\sigma_z^2)\log(n) \leq N$, $N\log(N) \leq n$ and $9d \leq N$, for some constant $c$ and with probability at least $1 - 2e^{-N/18} - 3n^{-10} - 3e^{-d} - e^{-n} - e^{-N}$. Inequality (ii) holds in the regime $(d + n\sigma_z^2)\log(n) \leq N$, which implies $\gamma < 1$ and we used the definition of $\tilde{\mathbf{D}}_{k2}$ from equation (22).

### G.2 PROOF OF EQUATION (10)

Recall that $\mathbf{D}_k = \text{diag}\left(\ldots, (1-(1-\eta\sigma_{y,i}^2)^k)/\sigma_{y,i}, \ldots\right) \in \mathbb{R}^{N\times N}$. The second term in equation (6) can be bounded as

$$\frac{\sigma_z^2}{d}\left\|\mathbf{X}\mathbf{V}_y\mathbf{D}_k\mathbf{U}_y^T\right\|_F^2 = \frac{\sigma_z^2}{d}\left\|\mathbf{U}_x\mathbf{\Sigma}_x\mathbf{V}_x^T\mathbf{V}_y\mathbf{D}_k\mathbf{U}_y^T\right\|_F^2$$

$$\leq\frac{\sigma_z^2}{d}\sigma_{x,max}^2\left\|\mathbf{V}_x^T\mathbf{V}_y\right\|^2\left\|\mathbf{D}_k\right\|_F^2$$

$$\leq\frac{\sigma_z^2}{d}\sum_{i=1}^{N}\frac{\sigma_{x,max}^2}{\sigma_{y,i}^2}(1-(1-\eta\sigma_{y,i}^2)^k)^2, \tag{32}$$

where we used that $\left\|\mathbf{V}_x^T\mathbf{V}_y\right\|^2 \leq 1$ and the definition of $\mathbf{D}_k$ from equation (22).

## H   AUXILIARY PROOFS

In this Section we provide a summary of auxiliary proofs that are used to prove the main results in Sections F and G.

### H.1   APPLYING THE SIN-THETA THEOREM TO BOUND THE DISTANCE BETWEEN SUBSPACES

In this Section, we use the following variant (Cai et al., 2015, Prop. 1) of the sin-theta theorem (Davis & Kahan, 1970) to bound the distances between subspaces occurring in the proofs in Section F and G.

**Proposition 1.** *Let* $\mathbf{Q}$ *and* $\hat{\mathbf{Q}}$ *be* $n \times n$ *symmetric matrices. Let* $r < n$ *be arbitrary and let* $\mathbf{U}$ *and* $\hat{\mathbf{U}}$ *be formed by the* $r$ *leading singular vectors of* $\mathbf{Q}$ *and* $\hat{\mathbf{Q}}$. *Then*

$$\left\| \hat{\mathbf{U}} \hat{\mathbf{U}}^T - \mathbf{U} \mathbf{U}^T \right\| \le \frac{2}{\sigma_r(\mathbf{Q}) - \sigma_{r+1}(\mathbf{Q})} \left\| \hat{\mathbf{Q}} - \mathbf{Q} \right\|. \tag{33}$$

Recall that $\mathbf{Y} = \mathbf{U}_y \mathbf{\Sigma}_y \mathbf{V}_y^T \in \mathbb{R}^{n \times N}$ and $\mathbf{X} = \mathbf{U}_x \mathbf{\Sigma}_x \mathbf{V}_x^T \in \mathbb{R}^{n \times N}$ are the SVDs of $\mathbf{Y}$ and $\mathbf{X}$ respectively. Let $\mathbf{U}_{y1}, \mathbf{U}_{x1} \in \mathbb{R}^{n \times d}$ be the $d$ leading left singular vectors of $\mathbf{Y}$ and $\mathbf{X}$, let $\mathbf{U}_{y2}, \mathbf{U}_{x2} \in \mathbb{R}^{n \times n-d}$ be the orthonormal complements. Let $\tilde{\mathbf{U}}_y = [\mathbf{U}_{y1} \quad \mathbf{U}_{y2}] \in \mathbb{R}^{n \times n}$ and $\tilde{\mathbf{U}}_x = [\mathbf{U}_{x1} \quad \mathbf{U}_{x2}] \in \mathbb{R}^{n \times n}$. Analogous definitions can be made for the leading right singular vectors of $\mathbf{Y}$ and $\mathbf{X}$.

We start by applying Proposition 1 to bound the distance between the subspaces spanned by the $d$ leading left singular vectors $\mathbf{U}_{y1}$ of $\mathbf{Y}$ and the subspace model $\mathbf{U}$ as

$$
\begin{aligned}
\left\| \mathbf{U}^T \mathbf{U}_{y2} \right\| &= \left\| \mathbf{U}_{y1} \mathbf{U}_{y1}^T - \mathbf{U} \mathbf{U}^T \right\| \\
&\overset{(i)}{\le} \left\| \frac{1}{N} \mathbf{Y} \mathbf{Y}^T - \mathbf{U} \mathbf{U}^T - \sigma_z^2 \mathbf{I} \right\| \\
&\overset{(ii)}{\le} \sqrt{\frac{c(d + n\sigma_z^2) \log(n)}{N}} + \frac{c(d + n\sigma_z^2) \log(n)}{N} \\
&\overset{(iii)}{\le} c\sqrt{\frac{(d + n\sigma_z^2) \log(n)}{N}},
\end{aligned} \tag{34}
$$

where inequality (i) follows from Proposition 1 and inequality (ii) holds with probability at least $1 - n^{-10} - 2e^{-d} + e^{-n}$ and follows from Section H.2 equation (45). Inequality (iii) holds in the regime $(d + n\sigma_z^2) \log(n) \le N$.

Next, we establish a bound for the distance between the subspaces spanned by the $d$ leading left singular vectors of $\mathbf{X}$ and $\mathbf{Y}$. We have that

$$
\begin{aligned}
\left\| \mathbf{U}_{y2}^T \mathbf{U}_{x1} \right\| &= \left\| \mathbf{U}_{y1} \mathbf{U}_{y1}^T - \mathbf{U}_{x1} \mathbf{U}_{x1}^T \right\| \\
&\le \left\| \mathbf{U}_{y1} \mathbf{U}_{y1}^T - \mathbf{U} \mathbf{U}^T \right\| + \left\| \mathbf{U}_{x1} \mathbf{U}_{x1}^T - \mathbf{U} \mathbf{U}^T \right\| \\
&\overset{(i)}{\le} \left\| \frac{1}{N} \mathbf{Y} \mathbf{Y}^T - \mathbf{U} \mathbf{U}^T - \sigma_z^2 \mathbf{I} \right\| + \left\| \frac{1}{N} \mathbf{X} \mathbf{X}^T - \mathbf{U} \mathbf{U}^T \right\| \\
&\overset{(ii)}{\le} \sqrt{\frac{c(d + n\sigma_z^2) \log(n)}{N}} + \frac{c(d + n\sigma_z^2) \log(n)}{N} + \sqrt{\frac{cd \log(n)}{N}} + \frac{cd \log(n)}{N} \\
&\le \sqrt{\frac{c(d + n\sigma_z^2) \log(n)}{N}} + \frac{c(d + n\sigma_z^2) \log(n)}{N} \\
&\overset{(iii)}{\le} c\sqrt{\frac{(d + n\sigma_z^2) \log(n)}{N}},
\end{aligned} \tag{35}
$$

where inequality (i) follows from Proposition 1 and inequality (ii) holds for some constant $c$ and with probability at least $1 - 2n^{-10} - 2e^{-d} + e^{-n}$ and follows from the results in Section H.2 equations (45),(46). Inequality (iii) holds in the regime $(d + n\sigma_z^2) \log(n) \le N$.

Last, we establish a bound for the distance between the subspaces spanned by the right singular vectors of $\mathbf{X}$ and $\mathbf{Y}$. We have that with probability at least $1 - 2e^{-N/18} - 2n^{-10} - e^{-n} - e^{-N} - e^{-d}$

$$
\begin{aligned}
\left\|\mathbf{V}_{x1}^T \mathbf{V}_{y2}\right\| &= \left\|\mathbf{V}_{x1}\mathbf{V}_{x1}^T - \mathbf{V}_{y1}\mathbf{V}_{y1}^T\right\| \\
&\overset{(i)}{\leq} \frac{2}{\sigma_{x,d}^2 - \sigma_{x,d+1}^2}\left\|\mathbf{Y}^T\mathbf{Y} - \mathbf{X}^T\mathbf{X}\right\| \\
&\overset{(ii)}{\leq} \frac{c}{N}\left\|\mathbf{Y}^T\mathbf{Y} - \mathbf{X}^T\mathbf{X}\right\| \\
&\leq c\left\|\frac{1}{N}\mathbf{Z}^T\mathbf{Z}\right\| + 2c\left\|\frac{1}{N}\mathbf{C}^T\mathbf{U}^T\mathbf{Z}\right\| \\
&\leq c\left\|\frac{1}{N}\mathbf{Z}\mathbf{Z}^T - \sigma_z^2\mathbf{I}\right\| + c\left\|\sigma_z^2\mathbf{I}\right\| + 2c\left\|\frac{1}{N}\mathbf{C}^T\mathbf{U}^T\mathbf{Z}\right\| \\
&\overset{(iii)}{\leq} \sqrt{\frac{cn\sigma_z^4\log(n)}{N}} + \frac{cn\sigma_z^2\log(n)}{N} + c\sigma_z\log(N) \\
&= c\sigma_z^2\sqrt{\frac{n\log(n)}{N}} + c\sigma_z^2\frac{n\log(n)}{N} + c\sqrt{\frac{\sigma_z^2 N\log(N)^2}{N}} \\
&\overset{(iv)}{\leq} c\sigma_z^2\frac{n\log(n)}{N} + c\sqrt{\frac{\sigma_z^2 n\log(n)}{N}} \\
&\overset{(v)}{\leq} c\sqrt{\frac{n\sigma_z^2\log(n)}{N}}.
\end{aligned}
\tag{36}
$$

Inequality (i) follows from Proposition 1. Inequality (ii) follows from the fact that $\sigma_{x,d+1}^2 = 0$ and that $\frac{1}{c}N \leq \sigma_{x,d}^2$ holds in the regime $9d \leq N$, with probability at least $1 - 2e^{-N/18}$ and for some constant $c$ (see Section H.4 equation (63)).
Inequality (iii) holds for some constant $c$ and with probability at least $1 - 2n^{-10} - e^{-n} - e^{-N} - e^{-d}$ and follows from the results in Section H.2 equations (47), (38). Inequality (iv) holds in the regime $N\log(N) \leq n$. Inequality (v) holds in the regime $n\sigma_z^2\log(n) \leq N$. To abbreviate notation from now on we define

$$
\psi := \frac{n\sigma_z^2\log(n)}{N}.
\tag{37}
$$

## H.2 Bounding a sum of independent random matrices

The Matrix Bernstein inequality (Oliveira, 2010; Tropp, 2012) can be used to bound a sum of independent, bounded and centered random matrices. We state the theorem below and then show how we applied it to bound several terms occurring in Sections F and G.

**Theorem 3** (Matrix Bernstein). *Let $\mathbf{S}_1, \ldots, \mathbf{S}_n$ be independent, centered random matrices with common dimension $d \times d$, and assume that each one is uniformly bounded*

$$
\mathbb{E}\left[\mathbf{S}_k\right] = \mathbf{0} \ \text{ and } \ \left\|\mathbf{S}_k\right\| \leq L \ \text{ for each } k = 1, \ldots, n.
$$

*Introduce the sum*

$$
\mathbf{Z} = \sum_{k=1}^{n}\mathbf{S}_k,
$$

*and let $v(\mathbf{Z})$ denote the matrix variance statistics of the sum:*

$$
\begin{aligned}
v(\mathbf{Z}) &= \max\left\{\left\|\mathbb{E}\left[\mathbf{Z}\mathbf{Z}^T\right]\right\|, \left\|\mathbb{E}\left[\mathbf{Z}^T\mathbf{Z}\right]\right\|\right\} \\
&= \max\left\{\left\|\sum_{k=1}^{n}\mathbb{E}\left[\mathbf{S}_k\mathbf{S}_k^T\right]\right\|, \left\|\sum_{k=1}^{n}\mathbb{E}\left[\mathbf{S}_k^T\mathbf{S}_k\right]\right\|\right\}.
\end{aligned}
$$

*Then with probability at least $1 - e^{-\delta}$ and $\delta \geq 0$*

$$
\|\mathbf{Z}\| \leq \frac{2}{3}L(\delta + \log(2d)) + \sqrt{2v(\mathbf{Z})(\delta + \log(2d))}.
$$

Recall our signal model to be $\mathbf{Y} = \mathbf{X} + \mathbf{Z} = \mathbf{U}\mathbf{C} + \mathbf{Z}$, with entries i.i.d. as $c_{i,j} \sim \mathcal{N}(0,1)$ and $z_{i,j} \sim \mathcal{N}(0, \sigma_z^2)$ and dimensions $\mathbf{Y}, \mathbf{X}, \mathbf{Z} \in \mathbb{R}^{n \times N}$, $\mathbf{C} \in \mathbb{R}^{d \times N}$. The subspace matrix $\mathbf{U} \in \mathbb{R}^{n \times d}$ has orthonormal columns.

We start by applying Theorem 3 to establish the following bound. With probability $1 - N^{-10} - e^{-N} - e^{-d}$ and for some unspecified numerical constant $c$ we have

$$\frac{1}{N} \left\| \mathbf{C}^T \mathbf{U}^T \mathbf{Z} \right\| \le c\sigma_z \log(N). \tag{38}$$

**Proof of equation** (38): We define $\tilde{\mathbf{Z}} = \mathbf{U}^T \mathbf{Z} \in \mathbb{R}^{d \times N}$. Note that the entries in $\tilde{\mathbf{Z}}$ are independent and identically distributed like the entries in $\mathbf{Z}$, i.e., $\tilde{z}_{i,j} \sim \mathcal{N}(0, \sigma_z^2)$. Next we check the conditions of applying Theorem 3 to bound $\left\| \mathbf{C}^T \tilde{\mathbf{Z}} \right\|$. Note that

$$\mathbf{C}^T \tilde{\mathbf{Z}} = \sum_{i=1}^{d} \mathbf{c}_i \tilde{\mathbf{z}}_i^T, \tag{39}$$

with $\mathbf{c}_i, \tilde{\mathbf{z}}[i] \in \mathbb{R}^N$ being the rows of $\mathbf{C}, \tilde{\mathbf{Z}}$. Since $\mathbf{c}_i$ has zero mean,

$$\mathbb{E}\left[ \mathbf{c}_i \tilde{\mathbf{z}}_i^T \right] = 0,$$

for all $i = 1, \ldots, d$. Further, we have

$$\begin{aligned}
\left\| \mathbf{c}_i \tilde{\mathbf{z}}_i^T \right\| &= \max_{\|\mathbf{w}\|_2 = 1} \left\| \mathbf{c}_i \tilde{\mathbf{z}}_i^T \mathbf{w} \right\|_2 \\
&= \max_{\|\mathbf{w}\|_2 = 1} |\tilde{\mathbf{z}}_i^T \mathbf{w}| \|\mathbf{c}_i\|_2 \\
&= \frac{\|\tilde{\mathbf{z}}_i\|_2^2}{\|\tilde{\mathbf{z}}_i\|_2} \|\mathbf{c}_i\|_2 \\
&= \|\tilde{\mathbf{z}}_i\|_2 \|\mathbf{c}_i\|_2 \\
&\le c\sigma_z N, 
\end{aligned} \tag{40}$$

where the inequality follows from equations (57),(56) and holds with probability at least $1 - e^{-N} - e^{-d}$. Finally we need to compute the matrix variance statistic $v(\mathbf{C}^T \tilde{\mathbf{Z}})$. Note that

$$\begin{aligned}
\mathbb{E}\left[ \mathbf{c}_i \tilde{\mathbf{z}}_i^T (\mathbf{c}_i \tilde{\mathbf{z}}_i^T)^T \right] &= \mathbb{E}\left[ \tilde{\mathbf{z}}_i^T \tilde{\mathbf{z}}_i \mathbf{c}_i \mathbf{c}_i^T \right] \\
&= \sigma_z^2 N \mathbf{I} \\
&= \mathbb{E}\left[ (\mathbf{c}_i \tilde{\mathbf{z}}_i^T)^T \mathbf{c}_i \tilde{\mathbf{z}}_i^T \right].
\end{aligned} \tag{41}$$

and therefore

$$\begin{aligned}
v(\mathbf{C}^T \tilde{\mathbf{Z}}) &= \max \left\{ \left\| \sum_{i=1}^{d} \mathbb{E}\left[ \mathbf{c}_i \tilde{\mathbf{z}}_i^T (\mathbf{c}_i \tilde{\mathbf{z}}_i^T)^T \right] \right\|, \left\| \sum_{i=1}^{d} \mathbb{E}\left[ (\mathbf{c}_i \tilde{\mathbf{z}}_i^T)^T \mathbf{c}_i \tilde{\mathbf{z}}_i^T \right] \right\| \right\} \\
&= \sigma_z^2 d N.
\end{aligned} \tag{42}$$

With equations (40),(42) in place we are ready to apply Theorem 3 to obtain with probability at least $1 - N^{-10} - e^{-N} - e^{-d}$ and some constant $c$

$$\begin{aligned}
\frac{1}{N} \left\| \mathbf{C}^T \mathbf{U}^T \mathbf{Z} \right\| &\le \sqrt{\frac{c\sigma_z^2 d \log(N)}{N}} + c\sigma_z \log(N) \\
&\le c\sigma_z \log(N),
\end{aligned} \tag{43}$$

where the last inequality holds since $d < N$. This concludes the proof of equation (38).

In the remainder of this Section we apply the example in Tropp (2015, Sec. 1.6.3) that illustrates how to use Theorem 3 to bound the distance between sample and true covariance matrices. With probability at least $1 - n^{-10}$ and an unspecified numerical constant $c$ we have

$$\left\| \frac{1}{N} \sum_{i=1}^{N} \mathbf{a}_i \mathbf{a}_i^T - \mathbb{E}\left[ \mathbf{a}\mathbf{a}^T \right] \right\| \le \sqrt{\frac{cB \|\mathbb{E}\left[ \mathbf{a}\mathbf{a}^T \right]\| \log(n)}{N}} + \frac{cB \log(n)}{N}, \tag{44}$$

where we assume that the $l_2$ norm of the random vector $\mathbf{a} \in \mathbb{R}^n$ is bounded $\|\mathbf{a}\|_2^2 \leq B$.

In the following we show how to apply (44) to establish the following bounds. With probability at least $1 - n^{-10} - 2e^{-d} + e^{-n}$ and for some unspecified numerical constant $c$

$$\left\| \frac{1}{N} \sum_{i=1}^{N} \mathbf{y}_i \mathbf{y}_i^T - \mathbb{E}\left[\mathbf{y}\mathbf{y}^T\right] \right\| = \left\| \frac{1}{N} \mathbf{Y}\mathbf{Y}^T - \mathbf{U}\mathbf{U}^T - \sigma_z^2 \mathbf{I} \right\|$$

$$\leq \sqrt{\frac{c(d + n\sigma_z^2)\log(n)}{N}} + \frac{c(d + n\sigma_z^2)\log(n)}{N}. \tag{45}$$

With probability at least $1 - n^{-10} - e^{-d}$ and for some unspecified numerical constant $c$

$$\left\| \frac{1}{N} \sum_{i=1}^{N} \mathbf{x}_i \mathbf{x}_i^T - \mathbb{E}\left[\mathbf{x}\mathbf{x}^T\right] \right\| = \left\| \frac{1}{N} \mathbf{X}\mathbf{X}^T - \mathbf{U}\mathbf{U}^T \right\|$$

$$\leq \sqrt{\frac{cd\log(n)}{N}} + \frac{cd\log(n)}{N}. \tag{46}$$

With probability at least $1 - n^{-10} - e^{-n}$ and for some unspecified numerical constant $c$

$$\left\| \frac{1}{N} \sum_{i=1}^{N} \mathbf{z}_i \mathbf{z}_i^T - \mathbb{E}\left[\mathbf{z}\mathbf{z}^T\right] \right\| = \left\| \frac{1}{N} \mathbf{Z}\mathbf{Z}^T - \sigma_z^2 \mathbf{I} \right\|$$

$$\leq \sqrt{\frac{cn\sigma_z^4\log(n)}{N}} + \frac{cn\sigma_z^2\log(n)}{N}. \tag{47}$$

In the notation of the general example in (44), the proofs of (45)-(47) consist of deriving expressions for $\|\mathbf{a}\|_2^2 \leq B$ and $\left\| \mathbb{E}\left[\mathbf{a}\mathbf{a}^T\right] \right\|$ respectively.

**Proof of equation (45):** We have

$$\|\mathbf{y}\|_2^2 = \mathbf{c}^T\mathbf{c} + 2\mathbf{z}^T\mathbf{U}\mathbf{c} + \mathbf{z}^T\mathbf{z}$$
$$\leq (5 + \sqrt{5})d + 5n\sigma_z^2$$
$$\leq c(d + n\sigma_z^2), \tag{48}$$

for some constant $c$ and where the first inequality holds with probability at least $1 - 2e^{-d} - e^{-n}$ as it is shown in Section H.3. Further, we have

$$\left\| \mathbb{E}\left[\mathbf{y}\mathbf{y}^T\right] \right\| = \left\| \mathbf{U}\mathbf{U}^T - \sigma_z^2\mathbf{I} \right\| = 1 + \sigma_z^2. \tag{49}$$

Since practical noise levels satisfy $\sigma_z^2 \leq 1$, there exists some constant $c$ such that

$$\|\mathbf{y}\|_2^2 \left\| \mathbb{E}\left[\mathbf{y}\mathbf{y}^T\right] \right\| \leq c(d + n\sigma_z^2), \tag{50}$$

with probability at least $1 - 2e^{-d} - e^{-n}$. Inserting (48) and (50) in the general form in (44) concludes the proof of equation (45).

**Proof of equation (46):** We have

$$\|\mathbf{x}\|_2^2 = \mathbf{c}^T\mathbf{c} \leq cd \tag{51}$$

for some constant $c$ and where the inequality holds with probability at least $1 - e^{-d}$ as it is shown in Section H.3. Further, we have

$$\left\| \mathbb{E}\left[\mathbf{x}\mathbf{x}^T\right] \right\| = \left\| \mathbf{U}\mathbf{U}^T \right\| = 1. \tag{52}$$

Inserting (51) and (52) in the general form in (44) concludes the proof of equation (46).

**Proof of equation (47):** We have

$$\|\mathbf{z}\|_2^2 \leq cn\sigma_z^2, \tag{53}$$

for some constant $c$ and where the inequality holds with probability at least $1 - e^{-n}$ as it is shown in Section H.3. Further, we have

$$\left\| \mathbb{E}\left[\mathbf{z}\mathbf{z}^T\right] \right\| = \sigma_z^2. \tag{54}$$

Inserting (53) and (54) in the general form in (44) concludes the proof of equation (47).

### H.3 Tail bounds for inner products of Gaussian vectors

Recall that $\mathbf{c} \sim \mathcal{N}(0, \mathbf{I}) \in \mathbb{R}^d$ and $\mathbf{z} \sim \mathcal{N}(0, \sigma_z^2 \mathbf{I}) \in \mathbb{R}^n$ are the columns of $\mathbf{C}, \mathbf{Z}$ respectively. Also $\mathbf{U} \in \mathbb{R}^{n \times d}$ has orthonormal columns. In this section we state some relations on the concentration of the inner products between those vectors. The chi-squared distributed $\mathbf{c}^T \mathbf{c}$ can be bounded with probability at most $e^{-t}$ as

$$\mathbf{c}^T \mathbf{c} \geq d + 2\sqrt{dt} + 2t. \tag{55}$$

Substituting $t$ with the signal dimension $d$ gives

$$\mathbf{c}^T \mathbf{c} \geq 5d, \tag{56}$$

with probability at most $e^{-d}$. Using the same result we can write

$$\mathbf{z}^T \mathbf{z} = \sigma_z^2 {\mathbf{z}'}^T \mathbf{z}' \geq 5n\sigma_z^2, \tag{57}$$

with probability at most $e^{-n}$ and where $\mathbf{z}' \sim \mathcal{N}(0, \mathbf{I})$. Next we show that with probability at most $2e^{-d}$

$$\mathbf{z}^T \mathbf{U} \mathbf{c} \geq \sqrt{5}d. \tag{58}$$

To this end, note that for any (deterministic) vector $\mathbf{c}$ we have $\mathbf{z}^T \mathbf{U} \mathbf{c} \sim \mathcal{N}(0, \|\mathbf{c}\|_2^2)$, and we apply a simple tail bound for Gaussian random variables to get

$$\mathbf{z}^T \mathbf{U} \mathbf{c} \geq \|\mathbf{c}\|_2 t, \tag{59}$$

with probability at most $e^{-t^2}$. It is straightforward to see that substituting $t$ with $\sqrt{d}$, applying (56) to bound $\|\mathbf{c}\|_2$ and combining the results with a union bound results into the bound (58). We can combine the results from this section to bound the sum

$$\mathbf{c}^T \mathbf{c} + 2\mathbf{z}^T \mathbf{U} \mathbf{c} + \mathbf{z}^T \mathbf{z} \geq (5 + \sqrt{5})d + 5n\sigma_z^2, \tag{60}$$

with probability at most $2e^{-d} + e^{-n}$.

### H.4 Bounding the extreme singular values of Gaussian random matrices and empirical covariance matrices

In this Section we state results for the extreme singular values of some of the matrices occurring in Sections F and G. Specifically, we establish equations (17),(18),(29) and (36) (ii).

A standard deviation inequality for the extreme singular values of some matrix $\mathbf{A} \in \mathbb{R}^{M \times m}$ with independent and identically standard normal distributed entries implies that (Rudelson & Vershynin, 2010, equation (2.3))

$$\sqrt{M} - \sqrt{m} - t \leq \sigma_{min}(\mathbf{A}) \leq \sigma_{max}(\mathbf{A}) \leq \sqrt{M} + \sqrt{m} + t, \tag{61}$$

with probability at least $1 - 2e^{-t^2/2}$ for $t \geq 0$.

We start with the largest singular value $\sigma_{x,max}$ of the feature matrix $\mathbf{X} \in \mathbb{R}^{n \times N}$. Recall that we have $\mathbf{X} = \mathbf{U}\mathbf{C}$ with orthonormal $\mathbf{U} \in \mathbb{R}^{n \times d}$ and $\mathbf{C} \in \mathbb{R}^{d \times N}$ with i.i.d. entries $c_{i,j} \sim \mathcal{N}(0, 1)$. Hence, the singular values of $\mathbf{U}\mathbf{C}$ are the singular values of $\mathbf{C}$.

Applying equation (61) to the matrix $\mathbf{C}$ under the assumption that the hidden signal dimension $d$ and the number of training examples $N$ fulfill $4d \leq N$ and choosing $t = \sqrt{N}/2$ we obtain

$$\begin{aligned}
\sigma_{x,max} &\leq \sqrt{N} + \sqrt{d} + \frac{\sqrt{N}}{2} \\
&\leq \sqrt{N} + \frac{\sqrt{N}}{2} + \frac{\sqrt{N}}{2} \\
&= 2\sqrt{N},
\end{aligned} \tag{62}$$

which holds with probability at least $1 - 2e^{-N/8}$. This establishes equation (17). For $\sigma_{x,min}$ the smallest non-zero singular value of $\mathbf{X}$ we have $\sigma_{x,min} = \sigma_{x,d} = \sigma_{c,d}$. In the regime $9d \leq N$ and choosing $t = \sqrt{N}/3$ we obtain

$$\sqrt{N} - \sqrt{d} - \frac{\sqrt{N}}{3} \leq \sigma_{x,min}$$

$$\sqrt{N} - \frac{\sqrt{N}}{3} - \frac{\sqrt{N}}{3} \leq \sigma_{x,min}$$

$$\frac{1}{c}\sqrt{N} \leq \sigma_{x,min}, \tag{63}$$

which holds with probability at least $1 - 2e^{-N/18}$ and for some constant $c$. This establishes (36) (ii).

Next, we state a bound on the largest singular value $\|\mathbf{Z}\|$ of the noise matrix $\mathbf{Z} \in \mathbb{R}^{n \times N}$ with i.i.d. entries $z_{i,j} \sim \mathcal{N}(0, \sigma_z^2)$. Note that $\mathbf{Z} = \sigma_z \tilde{\mathbf{Z}}$, where the entries of $\tilde{\mathbf{Z}}$ follow $\tilde{z}_{i,j} \sim \mathcal{N}(0, 1)$. Applying equation (61) with $t = \sqrt{N} + \sqrt{n}$ yields

$$\|\mathbf{Z}\| \leq c\sigma_z(\sqrt{N} + \sqrt{n})$$

$$\leq c\sigma_z\sqrt{n}, \tag{64}$$

with probability at least $1 - 2e^{-n/2}$ and some constant $c$. For the last inequality we assumed $n \geq N$. This establishes equation (29).

Finally, we derive bounds for some of the squared singular values of $\mathbf{Y} \in \mathbb{R}^{n \times N}$. In particular, we show that

$$\sigma_{y,d}^2 \geq (1 - \epsilon)N(1 + \sigma_z^2), \tag{65}$$

and

$$\sigma_{y,d+1}^2 \leq N\left(\sigma_z^2 + \epsilon(1 + \sigma_z^2)\right), \tag{66}$$

and finally

$$\sigma_{y,1}^2 \leq N\left((1 + \epsilon)(1 + \sigma_z^2)\right), \tag{67}$$

both hold with probability at least $1 - e^{-d} - e^{-n} - n^{-9}$ and for some constants $C$ and $\epsilon \in (0, 1)$ in the regime $N \geq 3C\epsilon^{-2}(d + \sigma_z^2 n)\log n$. This establishes equation (18).

To this end, we rely on the following Corollary (Vershynin, 2011, Corollary 5.52).

**Corollary 1** (Covariance estimation for arbitrary distributions). *Let $\mathbf{x}$ be a random vector in $\mathbb{R}^n$ supported in some centered Euclidean ball whose radius we denote $\sqrt{m}$. Consider $N$ independent samples $\mathbf{x}_i$ arranged as columns of the random matrix $\mathbf{A} \in \mathbb{R}^{n \times N}$. Denote $\Sigma_N = \frac{1}{N}\mathbf{A}\mathbf{A}^T$ as the sample covariance matrix and $\Sigma$ as the true covariance matrix. Let $\epsilon \in (0, 1)$ and $t \geq 1$. Then the following holds with probability at least $1 - n^{-t^2}$:*

$$\text{If } N \geq C(t/\epsilon)^2\|\Sigma\|^{-1}m\log n \text{ then } \|\Sigma_N - \Sigma\| \leq \epsilon\|\Sigma\|.$$

*Here C is an absolute constant.*

We apply Corollary 1 to the random vectors $\mathbf{y} = \mathbf{U}\mathbf{c} + \mathbf{z}$. Note that $\mathbb{E}\left[\mathbf{y}\mathbf{y}^T\right] = \mathbf{U}\mathbf{U}^T + \sigma_z^2\mathbf{I}$. Further note that $\mathbb{E}\left[\|\mathbf{y}\|_2^2\right] = d + \sigma_z^2 n$. We have

$$\|\mathbf{y}\|_2 \leq \|\mathbf{c}\|_2 + \|\mathbf{z}\|_2$$

$$\leq \sqrt{5d} + \sqrt{5n\sigma_z^2}$$

$$\leq c\sqrt{d + n\sigma_z^2}, \tag{68}$$

where the second inequality follows from Section H.3, (56), (57) and holds with probability at least $1 - e^{-d} - e^{-n}$.

Now we can apply Corollary 1 to make the following statement. For some constant $C$ and $\epsilon \in (0, 1)$ if $N \geq Ct\epsilon^{-2}(d + \sigma_z^2 n) \log n$, then with probability at least $1 - e^{-d} - e^{-n} - n^{-t^2}$

$$\left\| \frac{1}{N}\mathbf{Y}\mathbf{Y}^T - \mathbb{E}\left[\mathbf{y}\mathbf{y}^T\right] \right\| \leq \epsilon \|\mathbb{E}\left[\mathbf{y}\mathbf{y}^T\right]\| = \epsilon(1 + \sigma_z^2). \tag{69}$$

Consequently the singular values $\sigma_i(\frac{1}{N}\mathbf{Y}\mathbf{Y}^T)$ and $\sigma_i(\mathbb{E}\left[\mathbf{y}\mathbf{y}^T\right])$ differ by at most $\epsilon(1 + \sigma_z^2)$ and we can bound

$$\begin{aligned}
\sigma_{y,d}^2 &= N\sigma_d(\frac{1}{N}\mathbf{Y}\mathbf{Y}^T) \\
&\geq N(\sigma_d(\mathbb{E}\left[\mathbf{y}\mathbf{y}^T\right]) - \epsilon(1 + \sigma_z^2)) \\
&= N((1 + \sigma_z^2) - \epsilon(1 + \sigma_z^2)) \\
&= (1 - \epsilon)N(1 + \sigma_z^2).
\end{aligned} \tag{70}$$

and further

$$\begin{aligned}
\sigma_{y,d+1}^2 &= N\sigma_{y,d+1}\left(\frac{1}{N}\mathbf{Y}\mathbf{Y}^T\right) \\
&\leq N\left(\sigma_{y,d+1}\left(\mathbb{E}\left[\mathbf{y}\mathbf{y}^T\right]\right) + \epsilon(1 + \sigma_z^2)\right) \\
&= N\left(\sigma_z^2 + \epsilon(1 + \sigma_z^2)\right) \\
&= N\left((1 + \epsilon)\sigma_z^2 + \epsilon\right),
\end{aligned} \tag{71}$$

which concludes the proof of equations (65) and (66).

In the same manner we bound

$$\begin{aligned}
\sigma_{y,1}^2 &= N\sigma_{y,1}\left(\frac{1}{N}\mathbf{Y}\mathbf{Y}^T\right) \\
&\leq N\left(\sigma_{y,1}\left(\mathbb{E}\left[\mathbf{y}\mathbf{y}^T\right]\right) + \epsilon(1 + \sigma_z^2)\right) \\
&= N\left(1 + \sigma_z^2 + \epsilon(1 + \sigma_z^2)\right) \\
&= N\left((1 + \epsilon)(1 + \sigma_z^2)\right),
\end{aligned} \tag{72}$$

which concludes the proof of (67).

