# OpenReview forum: "Scaling Laws For Deep Learning Based Image Reconstruction"
_ICLR.cc/2023/Conference — ICLR 2023 poster_

### Official Review · Reviewer_nBGL · 2022-10-23

**Confidence:** 2
**Correctness:** 3
**Technical Novelty And Significance:** 3
**Empirical Novelty And Significance:** 3
**Recommendation:** 6

**Clarity, Quality, Novelty And Reproducibility:**

The overall flow of manuscript is well written and quality of writing is also fine.
The novelty of the work is appealing and authors submitted their code in supplementary.

**Strength And Weaknesses:**

(strength)
+ This paper analyzes the scalability of training dataset in linear inverse problems (denoising, super resolution, MRI acceleration). This aspect has never been analyzed significantly, so I believe this manuscript is first attempt to give a focus on the scalability of dataset in image restoration problem. This aspect is quite important in terms of intuitive instructions of building database.
Furthermore, the authors try to elaborate such the scalability property with rigorous theories which are supported in supplementaries in details.

(weakness)
- The way of writing from section 5 until conclusion is a bit distracted. This manuscript has a target to anlayze the data scalability, but their explanation from section 5 seems to be irrelevant with their claim. It would be easier  than the current. For example, in Fig. 3, W^\infty is not explained in caption and section, rather authors provide a link to supplementery which is not self-contained. I believe there is a need to be compact writing.

**Summary Of The Paper:**

This paper studied whether performance gains of neural network approaches for inverse/regression problems are obtained from scaling up the training set size. They reported their claims in three fields: image denoising, accelerated MRI, and super-resolution.
They calculated reconstruction quality as a function of training set size, while optimally scaling the network size. The authors found that an initial performance increasement slowly downs already at moderate training set sizes(~10^3). They confirmed that even training on millions of images would not significantly improve performance over small-sized training.
They concluded that once the error induced by learning the signal model is small relative to the error floor, more training examples do not improve performance.

**Summary Of The Review:**

The paper analyzes the scalability of training dataset in linear inverse problems and this is quite needed property from this society. I'm convinced of their claims and the authors prepared well-designed experiments to back up their claims. I believe their work is a good work. This paper will help relevant societies a lot.

---

> ### Author Response · Authors · 2022-11-14
> **Response to reviewer nBGL**
>
> We thank the reviewer for the feedback and for pointing out the novelty, importance of the results, and noting that the manuscript is well written.
>
> Regarding `the way of writing from section 5 until conclusion is a bit distracted': Thanks for noting that. We revised section 5 as well as section 6, to make the role of section 5 clearer, and to fix the issues pointed out by the reviews. For example, we now define $R(\mathbf{W}^\ast)$ in the main body, and have moved Figure 3 to the appendix, and added a new appendix (Appendix E) to explain the early-stopped estimator (former Fig. 3) better.
>
> The goal of section 5 is to study in a simple theoretical setup what risk behavior is expected. In section 5, we formalize, within the framework of a linear model, that as a function of training examples, performance first improves (with the improvement beeing approximated by a power law), and then enters a problem-specific error floor, where performance improvements as a function of training examples are marginal.
>
> We hope that our changes and the clarifications above address the reviewer's concerns and would be very happy if the reviewer would consider raising their score. We are also happy to discuss further, thanks again for your comments.

---

### Official Review · Reviewer_aCDc · 2022-10-25

**Confidence:** 4
**Correctness:** 3
**Technical Novelty And Significance:** 3
**Empirical Novelty And Significance:** 3
**Recommendation:** 5

**Clarity, Quality, Novelty And Reproducibility:**

- This work presented limited experimental results only using U-Net and SwinIR. The trends of these results look similar, but they are different in details. Thus, it will be also different for other networks. Can we say that there is a common phenomenon by looking into two different examples? More comprehensive experiments with a lot more networks may help to find this common aspect. Moreover, the most worried concern that I have is this question: "Are U-Net or SwinIR scalable?" It may be possible that this work simply investigated two non-scalable networks and there might be a scalable network for inverse problems. Another possible bad scenario is that the observed trends may not be the result of large training data, but the result of difficult global convergence for high dimensional spaces! Thus, in my view, it seems that this work may not be ready to draw the conclusion that the title and the conclusion claimed.

- The experiments in this work seem to be limited in many aspects besides the network types. This work covers the size of the training set up to 100K, but as mentioned above, there was a work using 1M samples (Brooks et al., 2019). It will be more convincing to cover larger settings than current settings to show really interesting areas to look into. Moreover, the way of increasing the size of networks is important (e.g., increasing depth, width and/or channel), but this work did not discuss this aspect at all. While it was claimed that the optimal parameter number was found for each case, but it is unclear if it is really optimal in terms of the way of increasing the network size. how about using EfficientNets since they are providing different network sizes?

**Strength And Weaknesses:**

Strengthes:
- Comprehensive empirical results are presented.

Weaknesses:
- Experiments seem to be limited to a very small set of all possible cases so it is unclear if the conclusion is well supported by the current experiments (see below for more details).
- It is unclear how the proposed conclusion can help researchers in inverse problems. Its usefulness and practical value are not really clear.
- The assumption on "Current methods are only trained on a few hundreds or thousands of images as opposed to the millions of examples deep networks are trained on in other domains" seems somewhat biased in my view. Tasks are different and the way of using one image is different (e.g., one image can not be divided into two or more independent images for a label in classification task, but it is common that multiple independent patches are extracted for training in denoising or super resolution tasks). Moreover, see (Brooks et al., 2019) that used 1M training samples for training a denoiser. Thus, it is hard to agree with the motivation of this work.
- The analysis with a linear estimation learned with early stopped gradient descent seems quite far from modern deep learning based methods that are highly non-linear and fully converged. Thus, all the conclusions made in this work seem over-claimed.

**Summary Of The Paper:**

This work investigated on the relationships between the performance in deep learning based inverse problems and the size of the training dataset. Denoising, compressive sensing MRI and super resolution problems were studied for U-Net and SwinIR. Theoretical analyses were also presented under simplified conditions. The conclusion of this work was that more training images beyond a certain point do not improve performance once the error induced by learning the signal model is small relative to the error floor, which was intuitively considered.

**Summary Of The Review:**

While this work attempted to investigate an interesting problem on the relationship between the performance of inverse problems in imaging and the size of training sets, the presented experiments seem limited and there are a number of potential issues in them. Thus, I can not recommend its acceptance in its current form.

---

> ### Author Response · Authors · 2022-11-14
> **Response to reviewer aCDc**
>
> Many thanks for the review. The comments lead us to justify the choices we made in our paper better. We also appreciate that the reviewer notes that "comprehensive empirical results are demonstrated", and that we "investigate an interesting problem". Below are responses to the concerns:
>
> * **Experiments seem to be limited to a small number of all possible cases:** Since every method has to be carefully tuned for each training set and network size, computing scaling laws for a large number of networks and setups is computationally infeasible. This is why scaling law papers focus on one or two architectures (e.g., Kaplan et al. on transformers and LSTMs, and Zhai et al. only on vision transformers). The scaling law curves alone (without counting fine tuning) for the two different architectures and the three different problems took 23k GPU hours. Adding the scaling law plot for MRI with the SwinIR transformer would cost 30k USD in AWS resources, and adding more networks and training on 10x more data as requested by the reviewer would cost hundreds of thousands of USD. Importantly, those experiments would likely yield limited insights, since we already study a CNN and a transformer, and all current state-of-the-art deep learning models for image reconstruction use one of the two.
>
> * **Usefulness and practical value of our results:** That is a great question. We started our work with the question: would it be beneficial to have a datasets resembling the fastMRI dataset but say 10x the size? Knowing the performance of deep learning based models for image reconstruction as a function of the number training examples is useful for assessing the cost and benefit of collecting such a large dataset. Our results indicate that only minor benefits in in-distribution performance are expected by scaling the training set size beyond a few thousand images. For denoising an architectural improvement like changing the architecture from a U-net to SwinIR, yields a much larger benefit than scaling up U-Net, for example.
>
> * **How to count training examples if images are split into patches:** The reviewer states that our statement "Current methods are only trained on a few hundreds or thousands of images as opposed to the millions of examples deep networks are trained on in other domains" is biased because ``tasks are different and the way of using one image is different''. We agree that tasks are different, but this statement is to explain that for other supervised machine learning problems, scaling the number of training examples turned out to be very beneficial for performance, and thus it is interesting to see whether scaling also significantly improves image reconstruction methods.
> We agree that there are different ways how to use an image for training (we focus on training on 256x256 sized patches), but the main question is whether scaling is beneficial and to what extent, which is what our paper addresses. We added ablation studies on the patch size to Appendix D.3.
>
> * **The size of our training sets being limited to 100k images:** We did not increase the training set size beyond 100k images, because the interesting phenomena (a significant slowing of the scaling law) occurred much earlier. We already get new state-of-the-art results on four widely used denoising datasets, showing that we clearly operate in an interesting regime.
> Regarding Brooks et al.: They train a much smaller U-net on 1M patches of size 128x128, on much less compute (72 GPU hours vs. 6700 GPU hours for our largest model). Our results indicate that a much smaller training set, but training on more epochs would have given the same or better results. Brooks at al. do not study the training set size. The only other work that we are aware of that trained on a similar sized training set (also about 1M images) is Chen et al.. As discussed in the paper, the resulting network is outperformed by our networks trained end-to-end on much smaller datasets.
>
> * **The theoretical analysis for a linear model being far from modern deep learning based methods, and ``all the results being overclaimed'':** We make no statements about neural networks in section 5, all statements in this section pertain to the linear model. We therefore respectfully disagree that any result in this section is overclaimed. The theory section is to provide qualitative predictions, and our results there describe the qualitative behavior of a model trained end-to-end to reconstruct a structured signal. Specifically, our theoretical analysis shows that once the error induced by learning the signal model is small relative to the error floor, more training examples do not improve performance, and that is also observed for the neural network based denoisers.

---

> > ### Author Response · Authors · 2022-11-14
> > **Response to reviewer aCDc (cont'd)**
> >
> > * **Scalability of the U-Net and SwinIR:** The reviewer is concerned that we `simply investigated two non-scalable networks and there might be a scalable network': By scaling up SwinIR, we get a new state-of-the-art result on four denoising datasets, on which hundreds of methods have been evaluated. This is a very strong indication that the methods are scalable.
> >
> > * **The observed trends possibly not being the result of large training data, but of `difficult global convergence for high dimensional spaces':** In Appendices A,C and D we describe how we ensure that the methods converge.
> >
> > * **Regarding the scaling up of network size:** We carefully investigated several ways to scale up the models, and choose the best one. For SwinIR we scaled up as suggested by the ablation studies of the original paper. For the U-Net we tried different numbers of down-/upsampling layers (depth), and number of input channels (width), and chose the best for each problem (e.g., depth 2 for denoising, 1 for super-resolution and 4 for MRI). Then we optimize over the network width as depicted in Fig. 1 b,d and Fig. 4 b.
> > However, even though we made an extensive effort to optimize hyper-parameters, we cannot know whether we found the optimal parameters, and made sure to never say `optimal' anywhere in the revised paper.
> >
> > We hope that our changes and the clarifications above address the reviewer's concerns and would appreciate it if the reviewer would consider raising their score. We are also happy to discuss further, thanks again for your comments.

---

### Official Review · Reviewer_ksGx · 2022-10-25

**Confidence:** 4
**Clarity, Quality, Novelty And Reproducibility:** The paper is well-written with modera…
**Correctness:** 2
**Technical Novelty And Significance:** 3
**Empirical Novelty And Significance:** 3
**Recommendation:** 5

**Strength And Weaknesses:**

Strengths:

1. The paper shows that deep learning-based imaging tasks does not require immense amount of data unlike other domains such as NLP.
2. The experimental results can navigate designing the amount of training data for imaging applications

Weaknesses:
1) Authors motivate their study by contrasting with another domain such as NLP. While it is true that imaging applications require far less data compared to NLP, authors do not really explain/motivate the reason behind it.  Authors try to explain the performance in terms of network size and training size. However, in imaging applications, every pixel in an image can be considered as a sample. Thus, the available amount of data is scaled exponentially.  Indeed, in many recent denoising applications, a set of pixels are used as a label rather than entire image (e.g. Noise2Void, Noise2Self, …). In imaging applications such as image denoising or reconstruction, a moderate performance can be achieved with even 1 sample/image (e.g., self2self). In this perspective, in imaging applications, available data for the task of interest is much larger than the number of images.

2) In denoising applications, quantitative results might be used as a metric to back-up the results. However, this cannot be the only metric for MRI reconstruction shown in the study. A high SSIM does not necessarily translates to an artifact-free reconstruction. A method can achieve a very high SSIM while suffering from artifacts (e.g., Muckley et al. 2021). Thus, in MRI or similar medical imaging applications, the amount of training data can be crucial in removing coherent artifacts which worsen with higher acceleration rates. Thus, I find the claims in section 4 questionable.  Extension to such domains could be a separate study that should be supported with extensive qualitative results.




**Summary Of The Paper:**

This paper investigates the impact of training set size for imaging applications. The paper shows that increasing training size will not yield similar gain in terms of performance. Quantitative experimental results conducted on image denoising and reconstruction aligns with the claims.

**Summary Of The Review:**

The paper investigates the impact of training data size. While this work might be useful for designing training data size for some imaging applications, I feel the approach proposed in this paper is not well-motivated. Moreover, generalizations of the results to different imaging applications are questionable since only some quantitative metrics which may not be reflective of the performance are provided.

---

> ### Author Response · Authors · 2022-11-14
> **Response to reviewer ksGx**
>
> Thanks for the feedback. In the following we address the weaknesses in the order as pointed out by the reviewer.
>
> 1.
>     1. **Regarding an explanation why applications in imaging require less data than in NLP:** Our intuition (without rigorous evidence) is that in NLP and perception tasks we have to understand the semantics of the data, while for image reconstruction, as we view it, we do not, we only need to learn a good prior for images.
>     2. **Regarding the question how to count the amount of training data:** The reviewer writes ``in imaging applications, every pixel in an image can be considered as a sample. Thus, the available amount of data is scaled exponentially.'' We agree that the amount of information in an image depends on the size of the image. In our experiments we fix the patch size to 256x256. In this setting, our findings remain unchanged regardless of plotting the performance as a function of the number of pixels in the training set or the number of patches of size 256x256 pixels. The patch size has an impact on performance; we added an abblation study to Appendix D.3 showing that training on $N$ many 256x256 patches performs better than training on $4N$ many 128x128 patches, for moderate to large $N$.
> We also agree that zero-shot methods such as self2self (and even classical methods such as BM3D that do not rely on any training data) can be used for denoising. Those approaches rely on making assumptions about images (such as self-similarity, etc). However, in this work we consider supervised learning.
>
> 2. **Regarding quantitative metrics:** The main critique of the reviewer is that basing our results on quantitative metrics might be problematic since quantitative metrics `may not be reflective of the performance'.
> We added two new figures to the Appendix (Figure 4 and 5) with qualitative results. The example reconstructions support our quantitative study. They show that the gains in image quality by increasing the training set size from 500 to 2500 images are larger than what we get by increasing the training set size from 10000 to 50000, which correlates with our quantitative results.
> Having added those results, our study is still based largely on quantitative results. We use the standard metrics (PSNR and SSIM) used in the vast majority of image reconstruction works and competitions. We agree that those metrics and quantitative metrics for image reconstructions in general have shortcomings. All common metrics are known to not fully reflect human perception. Yet, the vast majority of studies on image reconstruction and all major competitions that we are aware of, including the FastMRI competition, rely heavily on quantitative results, for a lack of better alternatives. Thus, the critique on reporting quantitative results is not a critique pertaining to this paper, but a general critique on drawing conclusions on the performance of image reconstruction methods based on quantitative results.
> We agree with the reviewer that the results in Muckley et al. show that a reconstruction can achieve a high SSIM and still contain artifacts. However, Muckley et al.'s results also show (see Figure 5, Muckley et al.) that in general radiologists award the best qualitative rank to models with the best quantitative SSIM score.
> The reconstruction in Figure 5 in the appendix that we added show that there are small details that are not reconstructed by any of our models. However, it is not clear if even more data could help or if these details are simply not there since the information is lost due to the large undersampling factor. The same question has already been raised in the first edition of the fastMRI challenge (Knoll et al. 2020a, Fig. 2), and has not been answered yet. There all methods failed to reconstruct a particular small detail in the image.
>
> We hope that adding the visual comparisons, and the fact that they support our conclusions convince the reviewer to raise their score.

---

### Official Review · Reviewer_pVJy · 2022-10-25

**Confidence:** 5
**Correctness:** 3
**Technical Novelty And Significance:** 3
**Empirical Novelty And Significance:** 4
**Recommendation:** 6

**Clarity, Quality, Novelty And Reproducibility:**

There is novelty in both the scaling empirical plots and the theoretical guarantees but it is not clear whether Theorem 1 has already been proved in the literature or not and the authros fail to discuss it.
The error bars in Fig 3 are hidden behind the line symbols, which are quite useless.


**Strength And Weaknesses:**

Strengths:
- Obtains some empirical results on the best accuracy that could be obtained for different numbers of training examples for two image reconstruction tasks.
- Obtains some theoretical guarantees for another image reconstruction task.

Weaknesses:
- The scaling of the transformer on the MRI reconstruction is missing
- The literature review of the theoretical guaranteed for SVD reconstruction is weak. The authors fail to discuss how the results from Theorems 1 and 2 compare to the already existing theoretical results from the literature.
- Theorem 2 is a very loose bound that has a rate worse than Theorem 1 because of the 1/sqrt(N) term plus (7+2 xi)R(W*). From Theorem 2 it would result that early stopping is worse than PCA, which is inconsistent with the experiments.


**Summary Of The Paper:**

The paper conducts an empirical study of how the image reconstruction accuracy changes with the number of training observations for low-level tasks such as image denoising and compressed sensing. For that, multiple models with different numbers of parameters are trained for each training set, and the best test accuracy is considered for the final plot. The plot reveals that the accuracy has an initial zone of high improvement after which a zone with more modest improvement is observed. The paper also obtains the scaling plot for a vision transformer for image denoising. Finally, the paper obtains some theoretical guarantees for a related problem: low-rank linear denoising.

**Summary Of The Review:**

The paper shows promise in obtaining scaling laws for denoising, but it is not ready for publication at this time.

---

> ### Author Response · Authors · 2022-11-14
> **Response to reviewer pVJy**
>
> Many thanks for the feedback and for appreciating the novelty in the empirical results and the theoretical guarantees. In the following we address the weaknesses in the order as pointed out by the reviewer.
>
> 1. **Regarding the lack of empirical results for MR reconstruction with a transformer based architecture:** We agree that the transformer based results would be interesting, but we decided to omit it, since the expected insights do not justify its cost. From our denoising experiment, the expected cost is 16800 GPU hours (about 30k USD on an AWS ml.p3.8xlarge instance). Our main result is that the performance gain in deep learning based image reconstruction tasks from increasing the training set size slows already at moderate training set sizes. For MR reconstruction we observe this result with CNN based networks. Combined with our finding that this also holds for image denoising with CNN and transformer based networks, it is expected that this continues to hold for transformer based MR reconstruction. Of course it would be great to verify this, but we can't justify the cost of the experiment, and hope that the reviewer understands.
>
> 2. **Regarding a literature review of the theoretical guarantees for SVD reconstruction missing:** Thanks for pointing this out, we added four references on analysing the SVD/PCA estimate to the main body (three were already cited in the appendix, we added an additional one).
>
> 3. **Regarding plotting $R(\mathbf{W}) - R(\mathbf{W}^\ast)$ instead of $R(\mathbf{W})$, and a log-plot vs a log-log plot:**
> Thanks for this comment, it led us to significantly revise the discussion in Section 5, in order to explain our choice to plot the risk $R(\mathbf{W})$ (instead of $R(\mathbf{W}) - R(\mathbf{W}^\ast)$) and to explain, why the results are plotted in a log-log plot.
> In short, Figure 3 in our revised version now contains both the risk minus the optimal risk $R(\mathbf{W}) - R(\mathbf{W}^\ast)$ as a function of $N$ as well as the risk $R(\mathbf{W})$ as a function of $N$. Throughout the paper we show the risk $R(\mathbf{W})$ as a function on $N$, since in practice we do not know the optimal risk $R(\mathbf{W}^\ast)$. We agree that scaling coefficients are obtained through linear fits at different regions of the performance curve in a log-log plot, and we have made this clear in the revised discussion. This is also common practice in empirical scaling law papers from other domains (Zhai et al., Kaplan et al.).
> With regards to log-log vs log: It has to be log-log, even $R(\mathbf{W}) - R(\mathbf{W}^\ast)$  is not linear in a log plot, only in a log-log plot.
>
> 4. **Regarding a comparison of the Theorems and their simulations in Section 5:** In our revised version we added a paragraph in Section 5 that connects the results of the simulations in Fig. 3 (see previous point) with our statements in Theorems 1 and 2.
>
> 5. **Regarding the strong assumption in Thm. 2; of the ambient dimension being larger than the number of training examples and the consistency of the estimator:** In our revised version we added Section E to the appendix, in which we discuss the assumptions and results of Thm. 2 in more detail; we couldn't do so in the main body for a lack of space.
> Amongst others, we added the following statements to address the reviewer's question:
> >"Assumption $N \log(N) \leq n$ means we are in the high-dimensional regime; we think this is somewhat closer to reality (for example for denoising a $512 \times 512$ image, this would require the number of training examples to be smaller than $250$k), but we can derive an analogous bound for the regime $N \log(N) \geq n$, where the number of training examples is larger than the ambient dimension.”
>
>     And further
>     >“the error bound does not converge to $R(\mathbf{W}^\ast)$ as the number of training examples, $N$, converges to infinity. This is probably an artifact of our analysis, but it is unclear, at least to us, how to derive a substantially tighter bound.”
>
> 6. **Regarding missing alpha values and error bars in Fig. 4:** Thanks for pointing out the missing alphas, we added them to the plot in Figure 3 (former Figure 4). The error bars were already in the plot, but they are so small that they are barely visible.
>
> Many thanks again for the review and the helpful comments. If our responses alleviate your concerns (which we hope), we would be happy if you could increase your score, to reflect this. Please let us know if we can clarify anything else.

---

### Official Review · Reviewer_duwb · 2022-10-25

**Confidence:** 4
**Correctness:** 4
**Technical Novelty And Significance:** 3
**Empirical Novelty And Significance:** 4
**Recommendation:** 8

**Clarity, Quality, Novelty And Reproducibility:**

Clear and novel study of high quality but limited coverage. Should be reproducible from the details given in the text and especially the appendix.

**Strength And Weaknesses:**

Strengths: The paper is very well written in an easy to read language. Contents are novel and interesting to the expected audience of the conference. Main findings are presented very clearly and supported by well-designed easy to digest figures, that would allow to grasp the main findings from the figures even without reading the text in depth.

New SOTA in some of the tasks come as a side product of scaling but are only given as a side remark in the main story.

Weaknesses: As with most experimental deep-learning studies the findings depend on hyperparameter settings that may have more or less dramatic effects on the results. The implicit assumption, that hyperparameter choices are in all cases in a suitable range cannot be supported without solid ablation studies. They are not given in the current paper and thus results presented need to be taken with a grain of salt.

A weakness of the paper is that experiments on image denoising do not vary noise levels. Thus, in the experiments it remains open, how the power laws change with different noise levels or models. This would be quite interesting in practice, as the selected noise level sigma = 25 is unusually high. It remains unclear (at least to this reviewer) how the found tipping points, where the steep power laws flatten considerably, change with noise level. This is quantitatively investigated in the theoretical part of the paper and shown in Figure 4. There it seems that lower noise levels correspond to smaller needed data set sizes. For image denoising this may be different, due to the typical signal distribution in Fourier space, where low-frequent structures have higher amplitudes than high-frequent ones and thus high noise levels ‘occlude’ high-frequent structures more than coarser ones. Lower noise levels would therefore allow to recover larger parts of the image signal and thus larger datasets may be needed before reaching the noise floor. Additional experiments would be needed to clarify this question.


**Summary Of The Paper:**

The paper studies scaling of performance of deep learning-based methods for different image reconstruction tasks with respect to available training set sizes. Tasks investigated are U-net and SwinIR transformer for image denoising (and super-resolution in the appendix), as well as U-net for MRI compressive sensing. Experiments are pertinent and convincing. Discussions for the found power laws provide suitable insight. In addition, a theoretical study on linear subspace estimation from noisy data is provided, supporting the empirical observations by explaining/reproducing their general behavior in this simplified setting.

**Summary Of The Review:**

Good paper delivering insight above performance. The paper would be stronger with a clearer message towards noise depencence of the findings.

---

> ### Author Response · Authors · 2022-11-14
> **Response to reviewer duwb**
>
> Many thanks for the comments. We really appreciate the strengths pointed out, such as our study being clear, novel and of high quality. In the following we address the weaknesses in the order as pointed out by the reviewer.
> 1. **Regarding the dependence on hyperparameter settings and the lack of ablation studies in the paper:** We agree that in order to experimentally obtain the optimal performance for a given network over a range of training sets and network sizes the choice of hyperparameters is very important.
> For each of the scaling laws, our approach to finding suitable hyperparameters is described in Appendices A,C and D. In particular, the learning rate scheduler described in appendices A.1 and C picks an initial learning rate in a suitable regime and adjusts the number of training epochs such that models are trained to convergence. By carrying out ablation studies for several combinations of training sets and network sizes we found that the learning rates picked by our scheme work as well well as those found by a costly grid search.
> For the scaling laws with the SwinIR for denoising (Figure 2) the computational costs of training large transformer based networks on a lot of data prevented a hyperparameter search other than adjusting the learning rate to ensure stable training. However, scaling up the network led to a new SOTA on several benchmark datasets, which suggests that hyperparameters were, albeit perhaps not optimal, picked from a suitable range.
> We also expanded Appendix D with two new simulations; one is on the patch size as a hyperparameter.
>
> 2. **Regarding the lack of varying noise levels for the denoising experiments:** We agree that it is important to know how the power law change with noise levels. Hence, we added results for a smaller noise level $\sigma_z=15$ (Appendix D.2). To the current date models for training set sizes up to 10k images have converged and we added the plot to the Appendix (see Fig. 8).
> We observe an improvement of about 1.3dB in PSNR, which is expected since the irreducible error decreases for smaller noise levels. We also observe that the scaling coefficient for the smaller noise level $\sigma_z = 15$ (i.e., $\alpha = 0.0027$) is slightly steeper than that for the larger noise level $\sigma_z = 25$ (i.e., $\alpha = 0.0019$). This coincides with the qualitative behavior of the curves for subspace denoising in Figure 3, from our theoretical investigation.

---

> > ### Comment · Reviewer_duwb · 2022-11-24
> > **Thanks for the rebuttal -- disagree with recommendation of reviewer aCDc**
> >
> > Thank you for the carefully prepared rebuttal, especially adding the study on lower noise level. I am still convinced, that this is a good paper worth being presented.
> >
> > By this, I clearly disagree with the Recommendation of reviewer aCDc (3: reject, not good enough). While I understand the concerns, that experimental results may not necessarily carry over to other situations, there is not a lot one can do about it. We cannot test ‘all’ cases – impossible! – but focus on some currently relevant ones. This is what the paper does. And it gives theoretic insight to some degree. Here, the admittedly limited theoretic findings are consistent with experiments. By combining the experiments and theory, the paper provides valuable insight – not providing ‘the’ solution, i.e. a mathematical prove how scaling behaves, but still valuable insight. Also, clearly, in case we are interested in a different scenario not handled here, we could test if a similar scaling behavior can be observed. However, the current paper tells us, that it may very well be, that expensive collecting and processing more and more data may not do the trick.

---

### Decision · Program_Chairs · 2023-01-20

**Decision:**

Accept: poster

**Justification For Why Not Higher Score:**

By its nature, the most important scaling results are empirical, and limited to the particular task studies. Thus their applicability is limited.

**Justification For Why Not Lower Score:**

It is valuable for the community to learn about case studies such as this one, and it can spur future research into methods with superior scaling.

**Metareview: Summary, Strengths And Weaknesses:**

The paper empirically derives scaling laws for three image reconstruction problems (inverse estimation, denoising, and super-resolution) and two neural network architectures (CNN and transformer). Additionally, there is some theoretical justification provided. The empirical finding that performance saturates rather quickly is useful to know, and might spur research towards overcoming this limitation.

**Note From Pc:**

if the above contains the word "oral" or "spotlight" please see: "oral" presentation means -> notable-top-5% and "spotlight" means -> notable-top-25%. As stated in our emails, we are disassociating presentation type from AC recommendations